# Stochastic Gradient Methods with Preconditioned Updates

## Abstract

This work considers non-convex finite sum minimization. There are a number of algorithms for such problems, but existing methods often work poorly when the problem is badly scaled and/or ill-conditioned, and a primary goal of this work is to introduce methods that alleviate this issue. Thus, here we include a preconditioner that is based upon Hutchinson's approach to approximating the diagonal of the Hessian, and couple it with several gradient based methods to give new 'scaled' algorithms: `Scaled SARAH` and `Scaled L-SVRG`. Theoretical complexity guarantees under smoothness assumptions are presented, and we prove linear convergence when both smoothness and the PL-condition is assumed. Because our adaptively scaled methods use approximate partial second order curvature information, they are better able to mitigate the impact of badly scaled problems, and this improved practical performance is demonstrated in the numerical experiments that are also presented in this work.

## 1 Introduction

This work considers the following, possibly nonconvex, finite-sum optimization problem:

$$\min_{w \in \mathbb{R}^d} \left\{ P(w) = \tfrac{1}{n} \sum_{i=1}^n f_i(w) \right\}, \tag{1}$$

where $w \in \mathbb{R}^d$ is the model/weight parameter and the loss functions $f_i : \mathbb{R}^d \to \mathbb{R} \ \forall i \in [n] := \{1 \ldots n\}$ are smooth and twice differentiable. Throughout this work it is assumed that (1) has an optimal solution, with a corresponding optimal value, denoted by $w^*$, and $P^* = P(w^*)$, respectively.

Problems of the form (1) cover a plethora of applications, including empirical risk minimization, deep learning, and supervised learning tasks such as regularized least squares or logistic regression (Shalev-Shwartz & Ben-David, 2014). This minimization problem can be difficult to solve, particularly when the number of training samples $n$, or problem dimension $d$, is large, or if the problem is nonconvex.

Stochastic Gradient Descent (`SGD`) is one of the most widely known methods for problem (1), and its origins date back to the 1950s with the work of Robbins & Monro (1951). The explosion of interest in machine learning has led to an immediate need for reliable and efficient algorithms for solving (1). Motivated by, and aiming to improve upon, vanilla `SGD`, many novel methods have already been developed for convex and/or strongly convex instances of (1), including `SAG`/`SAGA` (Le Roux et al., 2012; Defazio et al., 2014), `SDCA` (Shalev-Shwartz & Zhang, 2013), `SVRG` (Johnson & Zhang, 2013; Xiao & Zhang, 2014), `S2GD` (Konečný & Richtárik, 2017) and `SARAH` (Nguyen et al., 2017), to name just a few. In general, these methods are simple, have low per iteration computational costs, and are often able to find an $\varepsilon$-optimal solution to (1) quickly, when $\varepsilon > 0$ is not too small. However, they often have several hyper-parameters that can be difficult to tune, they can struggle when applied to ill-conditioned problems, and many iterations may be required to find a high accuracy solution.

Non-convex instances of the optimization problem (1) (for example, arising from deep neural networks (DNNS)) have been diverting the attention of researchers of late, and new algorithms are being developed to fill this gap (Ghadimi & Lan, 2013; Ghadimi et al., 2016; Lei et al., 2017; Li et al., 2021b). Of particular relevance to this work is the `PAGE` algorithm presented in Li et al. (2021a). The algorithm is conceptually simple, involving only one loop, and a small number of parameters, and can be applied to non-convex problems (1). The main update involves either a minibatch `SGD` direction, or the previous gradient with a small adjustment (similar to that in `SARAH` (Nguyen et al., 2017)). The `Loopless SVRG` (L-SVRG) method (Hofmann et al., 2015; Qian et al., 2021), is also of particular interest here. It is a simpler 'loopless' variant of `SVRG`, which, unlike for `PAGE`, involves

an unbiased estimator of the gradient, and it can be applied to non-convex instances of problem (1).

For problems that are poorly scaled and/or ill-conditioned, second order methods that incorporate curvature information, such as Newton or quasi-Newton methods (Dennis Jr. & Moré, 1977; Fletcher, 1987; Nocedal & Wright, 2006), can often outperform first order methods. Unfortunately, they can also be prohibitively expensive, in terms of both computational and storage costs. There are several works that have tried to reduce the potentially high cost of second order methods by using only approximate, or partial curvature information. Some of these stochastic second order, and quasi-Newton (Jahani et al., 2021a; 2020) methods have shown good practical performance for some machine learning problems, although, possibly due to the noise in the Hessian approximation, sometimes they perform similarly to first order variants.

An alternative approach to enhancing search directions is to use a *preconditioner*. There are several methods for problems of the form (1), which use what we call a 'first order preconditioner' — a preconditioner built using gradient information — including `Adagrad` (Duchi et al., 2011), RMSProp (Tieleman et al., 2012), and `Adam` (Kingma & Ba, 2015). `Adagrad` (Duchi et al., 2011) incorporates a diagonal preconditioner that is built using accumulated gradient information from the previous iterates. The preconditioner allows every component of the current gradient to be scaled adaptively, but it has the disadvantage that the elements of the preconditioner tend to grow rapidly as iterations progress, leading to a quickly decaying learning rate. A method that maintains the ability to adaptively scale elements of the gradient, but overcomes the drawback of a rapidly decreasing learning rate, is RMSProp. It does this by including a momentum parameter, $\beta_2$ in the update for the diagonal preconditioner. In particular, at each iteration the updated diagonal preconditioner is taken to be a convex combination (using a momentum parameter $\beta_2$) of the (square) of the previous preconditioner and the Hadamard product of the current gradient with itself. So, gradient information from all the previous iterates is included in the preconditioner, but there is a preference for more recent information. `Adam` (Kingma & Ba, 2015) combines the positive features of `Adagrad` and RMSProp, but it also uses a first moment estimate of the gradient, providing a kind of additional momentum. `Adam` preforms well in practice, and is among the most popular algorithms for DNN.

Recently, second order preconditioners that use approximate and/or partial curvature information have developed and studied. The approach in `AdaHessian` (Yao et al., 2020) was to use a diagonal preconditioner that was motivated by Hutchinson's approximation to the diagonal of the Hessian (Bekas et al., 2007), but that also stayed close to some of the approximations used in existing methods such as `Adam` (Kingma & Ba, 2015) and `Adagrad` (Duchi et al., 2011). Because of this, the approximation often differed markedly from the true diagonal of the Hessian, and therefore it did not always capture good enough curvature information to be helpful. The work `OASIS` (Jahani et al., 2021b), proposed a preconditioner that was closely based upon Hutchinson's approach, and provided a more accurate estimation of the diagonal of the Hessian, and correspondingly led to improved numerical behaviour in practice. The preconditioner presented in Jahani et al. (2021b) is adopted here.

## 1.1 NOTATION AND ASSUMPTIONS

Given a Positive Definite (PD) matrix $D \in \mathbb{R}^{d \times d}$, the weighted Euclidean norm is defined to be $\|x\|_D^2 = x^T D x$, where $x \in \mathbb{R}^d$. The symbol $\odot$ denotes the Hadamard product, and `diag(x)` denotes the $d \times d$ diagonal matrix whose diagonal entries are the components of the vector $x \in \mathbb{R}^d$.

Recall that problem (1) is assumed to have an optimal (probably not a unique) solution $w^*$, with corresponding optimal value $P^* = P(w^*)$. As is standard for stochastic algorithms, the convergence guarantees presented in this work will develop a bound on the number of iterations $T$, required to push the expected squared norm of the gradient below some error tolerance $\varepsilon > 0$, i.e., to find a $\hat{w}_T$ satisfying

$$\mathbb{E}[\|\nabla P(\hat{w}_T)\|_2^2] \leq \varepsilon^2. \tag{2}$$

A point $\hat{w}_T$ satisfying (2) is referred to as an $\varepsilon$-optimal solution. Importantly, $\hat{w}_T$ is *some* iterate generated in the first $T$ iterations of each algorithm, but it is *not necessarily* the $T$th iterate.

Throughout this work we assume that each $f_i : \mathbb{R}^d \to \mathbb{R}$ and $P : \mathbb{R}^d \to \mathbb{R}$ are twice differentiable and also $L$-smooth. This is formalized in the following assumption.

**Assumption 1.1** ($L$-smoothness). For all $i \in [n]$ $f_i$ and $P$ are assumed to be twice differentiable and $L$-smooth, i.e., $\forall i \in [n], \forall w, w' \in \text{dom}(f_i)$ we have $\|\nabla f_i(w) - \nabla f_i(w')\| \leq L\|w - w'\|$, and $\forall w, w' \in \text{dom}(P)$ we have $\|\nabla P(w) - \nabla P(w')\| \leq L\|w - w'\|$.

For some of the results in this work, it will also be assumed that the function $P$ satisfies the PL-condition. Note that the PL-condition does not imply convexity (see Footnote 1 in Li et al. (2021a)).

**Assumption 1.2** (Polyak-Łojasiewicz-condition). A function $P : \mathbb{R}^d \to \mathbb{R}$ satisfies the PL-condition if there exists $\mu > 0$, such that $\|\nabla P(w)\|^2 \geq 2\mu(P(w) - P^*)$, $\forall w \in \mathbb{R}^d$.

## 1.2 CONTRIBUTIONS

The main contributions of this work are stated below, and are summarized in Tables 1 and 2.

• **Scaled SARAH.** We present a new algorithm called Scaled SARAH, which is a combination of the SARAH (Nguyen et al., 2017) and PAGE (Li et al., 2021a) algorithms, coupled with the diagonal preconditioner from Jahani et al. (2021b). The inclusion of the preconditioner results in adaptive scaling of every element of the search direction (negative gradient), which leads to improved practical performance, particularly on ill-conditioned and poorly scaled problems. The algorithm is simple (a single loop) and is easy to tune (Section 3).

• **Scaled L-SVRG.** The Scaled L-SVRG algorithm is also presented, which is similar to L-SVRG, but with the

Table 1: Comparison of scaled methods for non-convex problems. *Notation:* $\varepsilon$ denotes solution accuracy (2). The 'Tuning of $\beta_2$' column shows whether it is easy ('+'), or difficult ('−') to tune $\beta_2$. Our preconditioner (5) works with any $\beta_2$. Adam only supports certain large $\beta \approx 1$ (Défossez et al., 2020; Reddi et al., 2019).

| Method | Reference | Convergence | Tuning of $\beta_2$ |
|---|---|---|---|
| Adagrad | (Duchi et al., 2011) (Zou et al., 2018) (Défossez et al., 2020) | $\varepsilon^{-4}$ | + |
| RMSProp | (Tieleman et al., 2012) | no theory | |
| Adam | (Kingma & Ba, 2015) (Défossez et al., 2020) | $\varepsilon^{-4}$ | − |
| AdaHessian | (Yao et al., 2020) | no theory | |
| OASIS | (Jahani et al., 2021b) | $\varepsilon^{-4}$ | + |
| Scaled SARAH | This work | $\varepsilon^{-2}$ | + |
| Scaled L-SVRG | This work | $\varepsilon^{-2}$ | + |

Table 2: A summary of the main results of this work. Complexities for Scaled SARAH and Scaled L-SVRG are given for non-convex problems (first row), and under the PL assumption (second row). *Notation:* $L$ = smoothness constant, $\mu$ = PL constant, $\varepsilon$ = solution accuracy (2), $\Delta_0 = P(w^0) - P^*$, $n$ = data size, and $\Gamma, \alpha$ are upper and lower bounds of the Hessian approximation.

| | Scaled SARAH | Scaled L-SVRG |
|---|---|---|
| NC | $\mathcal{O}\left(n + \frac{\Gamma}{\alpha} \frac{\sqrt{n}L\Delta_0}{\varepsilon^2}\right)$ | $\mathcal{O}\left(n + \frac{\Gamma}{\alpha} \frac{n^{2/3}L\Delta_0}{\varepsilon^2}\right)$ |
| PL | $\mathcal{O}\left(\max\left\{n, \frac{\Gamma}{\alpha}\sqrt{n}\frac{L}{\mu}\right\} \log\frac{\Delta_0}{\varepsilon}\right)$ | $\mathcal{O}\left(\max\left\{n, \frac{\Gamma}{\alpha}n^{2/3}\frac{L}{\mu}\right\} \log\frac{\Delta_0}{\varepsilon}\right)$ |

addition of the diagonal preconditioner (Jahani et al., 2021b). Again, the preconditioner allows all elements of the gradient to be scaled adaptively, the algorithm uses a single loop structure, and for this algorithm an unbiased estimate of the gradient is used. The inclusion of adaptive local curvature information via the preconditioner leads to improvments in practical performance (Appendix A).

• **Convergence guarantees.** Theoretical guarantees show that both Scaled SARAH and Scaled L-SVRG converge and we present an explicit bound for the number of iterations required by each algorithm to obtain an iterate that is $\epsilon$-optimal. Convergence is guaranteed for both Scaled SARAH and Scaled L-SVRG under a smoothness assumption on the functions $f_i$. If both smoothness and the PL-condition hold, then improved iteration complexity results for Scaled SARAH and Scaled L-SVRG are obtained, which show that expected function value gap converges to zero at a linear rate (see Theorems 3.2 and A.2.) Our scaled methods achieve the best known rates of all methods with preconditioning for non-convex deterministic and stochastic problems, and Scaled SARAH and Scaled L-SVRG are the first preconditioned methods that achieve a linear rate of convergence under the PL assumption. See a detailed comparison in Section 4.

• **Numerical experiments.** Extensive numerical experiments were performed (Sections 5 and B) under various parameter settings to investigate the practical behaviour of our new scaled algorithms. The inclusion of preconditioning in Scaled SARAH and Scaled L-SVRG led to improvements in performance compared with no preconditioning in several of the experiments, and Scaled SARAH and Scaled L-SVRG were competitive with, and often outperformed, Adam.

**Paper outline.** This paper is organised as follows. In Section 2 we describe the diagonal preconditioner that will be used in this work. In Section 3, we describe a new Scaled SARAH algorithm and present theoretical convergence guarantees. In Section 4, we give discussions of our results for the Scaled SARAH method, and also compare it with other methods. In Appendix A, we introduce the Scaled L-SVRG algorithm, which adapts the L-SVRG algorithm to include a preconditioner.

We present numerical experiments demonstrating the practical performance of our proposed methods in Section 5. All proofs (Sections E and F), additional numerical experiments (Section B), and further details and discussion can be found in the appendix.

## 2 DIAGONAL PRECONDITIONER

In this section we describe the diagonal preconditioner that is used in this work. The paper of Bekas et al. (2007) described Hutchinson's approximation to the diagonal of the Hessian, and this provided motivation for the diagonal preconditioner proposed in Jahani et al. (2021b), which is adopted here. In particular, given an initial approximation $D_0$, (to be described soon), and Hessian approximation momentum parameter $\beta \in (0,1)$ (equivalent to the second moment hyperparameter, $\beta_2$ in `Adam` (Kingma & Ba, 2015)), for all $t \geq 1$,

$$D_t = \beta D_{t-1} + (1 - \beta)\texttt{diag}\left(z_t \odot \nabla^2 P_{\mathcal{J}_t}(w_t)z_t\right), \tag{3}$$

where $z_t$ is a random vector with Rademacher distribution[1], $\mathcal{J}_t$ is an index set randomly sampled from $[n]$, and

$$\nabla^2 P_{\mathcal{J}_t}(w_t) = \frac{1}{|\mathcal{J}_t|}\sum_{j \in \mathcal{J}_t} \nabla^2 f_j(w_t). \tag{4}$$

Finally, for $\alpha > 0$ (where the parameter $\alpha > 0$ is equivalent to the parameter $\epsilon$ in `Adam` (Kingma & Ba, 2015) and `AdaHessian` (Yao et al., 2020)), the diagonal preconditioner is:

$$\left(\hat{D}_t\right)_{i,i} = \max\{\alpha, |D_t|_{i,i}\}. \tag{5}$$

The expression (5) ensures that the preconditioner $\hat{D}_t$ is always PD, so it is well-defined and results in a descent direction. The absolute values are necessary because the objective function is potentially nonconvex, so the batch Hessian approximation could be indefinite. In fact, even if the Hessian is PD, $D_t$ in (3) may still contain negative elements due to the sampling strategy used.

The preconditioner (5) is a good estimate of the diagonal of the (batch) Hessian because Hutchinson's updating formula (3) is used (see also Figures 12, 13). Hence, it captures accurate curvature information, which is helpful for poorly scaled and ill-conditioned problems. Because the preconditioner is diagonal it is easy and inexpensive to apply it's inverse, and the associated storage costs are low.

The preconditioner (3)+(5) depends on the parameter $\beta$: if $\beta = 1$ then the preconditioner is fixed for all iterations, whereas if $\beta = 0$ then the preconditioner is simply a kind of sketched batch Hessian. Taking $0 < \beta < 1$ gives a convex combination of the previous approximation and the current approximation, thereby ensuring that the entire history is included in the preconditioner, but is damped by $\beta$, and the most recent information is also present. See Section G for a detailed discussion of the choice of $\beta$.

The main computational cost of the approximation in (3) is the (batch) Hessian-vector product $\nabla^2 P_{\mathcal{J}_t}(w_t)z_t$. Fortunately, this can be efficiently calculated using two rounds of back propagation. Moreover, the preconditioner is matrix-free, simply needing an oracle to return the Hessian vector product, but it does not need explicit access to the batch Hessian itself; see Appendix B in Jahani et al. (2021b). Therefore, the costs (both computational and storage) for this preconditioner are not burdensome.

As previously mentioned, the approximation (3) requires an initial estimate $D_0$ of the diagonal of the Hessian, and this is critical to the success of the preconditioner. In particular, one must take

$$D_0 = \frac{1}{m}\sum_{j=1}^{m}\texttt{diag}\left(z_j \odot \nabla^2 P_{\mathcal{J}_j}(w_0)z_j\right), \tag{6}$$

where $\mathcal{J}_j$ denotes sampled batches and the vectors $z_j$ are generated from a Radermacher distribution. This ensures that $\hat{D}_t$ does indeed approximate the diagonal of the Hessian; see Section 3.3 in Jahani et al. (2021b).

The following remark confirms that the diagonal preconditioner is both PD and bounded.

**Lemma 2.1** (See Remark 4.10 in Jahani et al. (2021b)). *For any $t \geq 1$, we have $\alpha I \preccurlyeq \hat{D}_t \preccurlyeq \Gamma I$, where $0 < \alpha \leq \Gamma = \sqrt{d}L$.*

---

[1]i.e., the components of the $z_t$ are $\pm 1$ with equal probability.

Note that Remark 4.10 is proved incorrectly in Jahani et al. (2021b).

## 3  Scaled SARAH

Here we propose a new algorithm, Scaled SARAH, for finite sum optimization (1). Our algorithm is similar to the SARAH algorithm (Nguyen et al., 2017) and the PAGE algorithm (Li et al., 2021a), but a key difference is that Scaled SARAH includes the option of a preconditioner, $\hat{D}_t$ for all $t \geq 0$, with a preconditioned approximate gradient step. Scaled SARAH is presented now as Algorithm 1.

---
**Algorithm 1** Scaled SARAH

---
1: **Input:** initial point $w_0$, learning rate $\eta$, preconditioner $\hat{D}_0$, probability $p$
2: $v_0 = \nabla P(w_0)$
3: **for** $t = 0, 1, 2, \ldots$ **do**
4:      $w_{t+1} = w_t - \eta \hat{D}_t^{-1} v_t$
5:      Generate independently batches $i_{t+1}$ for $v_{t+1}$ and $\mathcal{J}_t$ for $\hat{D}_{t+1}$
6:      $v_{t+1} = \begin{cases} \nabla P(w_{t+1}), & \text{with probability } p \\ v_t + \nabla f_{i_{t+1}}(w_{t+1}) - \nabla f_{i_{t+1}}(w_t), & \text{with probability } 1 - p \end{cases}$
7:      Update the preconditioner $\hat{D}_{t+1}$
8: **end for**
9: **Output:** $\hat{w}_T$ chosen uniformly from $\{w_t\}_{t=0}^T$

---

In each iteration of Algorithm 1 an update is computed in Step 4. The point $w_t$ is adjusted by taking a step in the direction $\hat{D}_t^{-1} v_t$, of fixed step size $\eta$. The vector $v_t$ approximates the gradient, and the preconditioner scales that direction. A key difference between Scaled SARAH and PAGE/SARAH is the inclusion of the preconditioner $\hat{D}_t^{-1}$ in this step.

Step 6 defines the next gradient estimator $v_{t+1}$, for which there are two options. With probability $p$ the full gradient is used. Alternately, with probability $1 - p$, the new gradient estimate is the previous gradient approximation $v_t$, with an adjustment term that involves the difference between the gradient of $f_i$ evaluated at $w_{t+1}$ and at $w_t$. The search direction computed in Scaled SARAH contains gradient information, while the preconditioner described in Section 2 contains approximate second order information. When this preconditioner is applied to the gradient estimate, each dimension is scaled adaptively depending on the corresponding curvature. Intuitively, this amplifies dimensions with low curvature (shallow loss surfaces), while damping directions with high curvature (sharper loss surfaces). The aim is for $\hat{D}_t^{-1} v_t$ to point in a better, adjusted direction, compared with $v_t$.

Scaled SARAH is a single loop algorithm so it is conceptually simple. If $p = 1$ then the algorithm always picks the first option in Step 6, so that Scaled SARAH reduces to a preconditioned GD method. On the other hand, if $p = 0$, then only the second option in Step 6 is used.

Notice that Scaled SARAH is a combination of both the PAGE and SARAH algorithms, coupled with a preconditioner. SARAH (Nguyen et al., 2017) is a double loop algorithm, where the inner loop is defined in the same way as update in Step 6. PAGE (Li et al., 2021a) is based upon SARAH, but PAGE uses a single loop structure, and allows for minibatches to be used in the gradient approximation $v_{t+1}$ (rather than the single component as in Step 6).[2] Scaled SARAH shares the same single loop structure as PAGE, but also shares the same single component update for the gradient estimator as SARAH (no minibatches). However, different from both PAGE and SARAH, Scaled SARAH uses a preconditioner in Step 4.

In the remainder of this work we focus on a particular instance of Scaled SARAH, which uses a fixed probability $p_t = p$, and uses the diagonal preconditioner presented in Section 2. These choices have been made because a central goal of this work is to understand the impact that a well chosen preconditioner has on poorly scaled problems. Convergence guarantees and the results of numerical experiments, will be presented using this set up.

Theoretical results for Scaled SARAH are presented now. In particular, we present complexity bounds on the number of iterations required by Scaled SARAH to obtain an $\varepsilon$-optimal solution for the non-convex problem (1) (recall Section 1.1 and (2)). The first result holds under Assumption 1.1,

---

[2]Note that, while PAGE allows minibatches for either option in the update Step 6, most of the theoretical results presented in Li et al. (2021a) require the full gradient to be computed as the first option in Step 6.

while the second theorem holds under both smoothness *and* PL assumptions. First, we define the following step-size bound:

$$\bar{\eta} = \frac{\alpha}{L\left(1 + \sqrt{\frac{1-p}{p}}\right)}. \tag{7}$$

**Theorem 3.1.** *Suppose that Assumption 1.1 holds, let $\varepsilon > 0$, let $p$ denote the probability, and let the step-size satisfy $\eta \leq \bar{\eta}$ (7). Then, the number of iterations performed by* `Scaled SARAH`, *starting from an initial point $w_0 \in \mathbb{R}^d$ with $\Delta_0 = P(w_0) - P^*$, required to obtain an $\varepsilon$-approximate solution of the non-convex finite-sum problem* (1) *can be bounded by*

$$T = \mathcal{O}\left(\frac{\Gamma}{\alpha}\frac{\Delta_0 L}{\varepsilon^2}\left(1 + \sqrt{\frac{1-p}{p}}\right)\right).$$

**Theorem 3.2.** *Suppose that Assumptions 1.1 and 1.2 hold, let $\varepsilon > 0$, and let the step-size satisfy $\eta \leq \bar{\eta}$ (7). Then the number of iterations performed by* `Scaled SARAH` *sufficient for finding an $\varepsilon$-approximate solution of non-convex finite-sum problem* (1) *can be bounded by*

$$T = \mathcal{O}\left(\max\left\{\frac{1}{p}, \frac{L}{\mu}\frac{\Gamma}{\alpha}\left(1 + \sqrt{\frac{1-p}{p}}\right)\right\}\log\frac{\Delta_0}{\varepsilon}\right).$$

Note that this last theorem shows that `Scaled SARAH` exhibits a linear rate of convergence under both the smoothness assumption and the PL-condition.

We know that Algorithm 1 calls the full gradient at the beginning (Step 2) and then uses $pn + (1 - p)$ stochastic gradients for each iteration on expectation (Step 6). Thus, the number of stochastic gradient computations (i.e., gradient complexity) is $n + T[pn + (1 - p)]$ and the following corollaries of Theorems 3.1 and 3.2 are valid.

**Corollary 3.3.** *Suppose that Assumption 1.1 holds, let $\varepsilon > 0$, let $p = \frac{1}{n+1}$, and let the step-size satisfy $\eta \leq \bar{\eta}$ (7). Then, the stochastic gradient complexity performed by* `Scaled SARAH`, *starting from an initial point $w_0 \in \mathbb{R}^d$ with $\Delta_0 = P(w_0) - P^*$, required to obtain an $\varepsilon$-approximate solution of the non-convex finite-sum problem* (1) *can be bounded by $\mathcal{O}\left(n + \frac{\Gamma}{\alpha}\frac{\Delta_0 L}{\varepsilon^2}\sqrt{n}\right)$.*

**Corollary 3.4.** *Suppose that Assumptions 1.1 and 1.2 hold, let $\varepsilon > 0$, and let the step-size satisfy $\eta \leq \bar{\eta}$ (7). Then the stochastic gradient complexity performed by* `Scaled SARAH` *sufficient for finding an $\varepsilon$-approximate solution of non-convex finite-sum problem* (1) *can be bounded by $\mathcal{O}\left(\left\{n + \frac{L}{\mu}\frac{\Gamma}{\alpha}\sqrt{n}\right\}\log\frac{\Delta_0}{\varepsilon}\right)$.*

In Appendix A we give our second method `Scaled L-SVRG`, it has worse convergence results than `Scaled SARAH`, but is be investigated in experiments, but is not be mentioned in the next section with discussions and comparison with competitors results.

## 4 DISCUSSION

In this section, we discuss the obtained result for `Scaled SARAH`, including how it relates to the same results for other scaled methods as well as for methods without preconditioning. For convenience, we give Table 3 summarising all the results.

• Despite in Section 2 we consider scaling based on the Hutchinson's approximation, but with our analysis one can obtain similar estimates for `Scaled SARAH` with `Adam` preconditioning. In particular, we can prove an analogue of Lemma 2.1 (see Appendix D) by additionally assuming boundedness of the stochastic gradient for all $w$: $\|\nabla f_i(w)\| \leq M$ (a similar assumption is made in Défossez et al. (2020)). We present these results in Table 3 for comparison with the current best results for `Adam`.

• In the deterministic case our results are significantly superior to those from Défossez et al. (2020), in particular, in terms of the accuracy of the solution our estimates give $\mathcal{O}(\varepsilon^{-2})$ dependence, at the same time the guarantees from Défossez et al. (2020) are $\mathcal{O}(\varepsilon^{-4})$. Compared to OASIS in the deterministic case we have the same results in terms of $\varepsilon$, but our bounds are much better in terms of $d$, $L$, $\alpha$. It is also an interesting detail that our estimates for `Scaled SARAH` with Adam preconditioner are independent of $d$ and with Hutchinson's preconditioner depend on $\sqrt{d}$, which is important for high-dimensional problems.

• In the stochastic case, our convergence guarantees are also the best among other scaled methods primarily in terms of $\varepsilon$. This is mainly due to the fact that we use the stochastic finite sum setting typical for machine learning.

• Unfortunately, our estimates are inferior to bounds of the unscaled methods: SARAH (the base method for our method) and SGD (the best known method for minimization problems). As one can see in Table 3, all results for methods with preconditioning have the same problem. This is the level of theory development in this field at the moment. It seems that our results are able in some sense to reduce this gap between scaled and unscaled methods by decreasing the additional multiplier.

Table 3: Comparison of deterministic and stochastic methods for non-convex problems in the general case and under Polyak-Łojasiewicz condition. In the stochastic case, the table is divided into two parts: the bounded and finite-sum setups. *Notation:* $\sigma^2$ = variance of stochastic gradients, $M$=uniform bound of (stochastic) gradients, the rest of the notation is the same as the one introduced earlier in the paper.

| | Method and reference | Non-convex | Polyak-Łojasiewicz |
|---|---|---|---|
| **Deterministic** | SGD (Secs. B.2 and C.2 from Li & Richtárik (2020)) | $\mathcal{O}\left(\frac{L\Delta_0}{\varepsilon^2}\right)$ | $\mathcal{O}\left(\frac{L}{\mu}\log\frac{1}{\varepsilon}\right)$ |
| | SARAH (Sec. 3.2 from Pham et al. (2020), Sec. 5 from Li et al. (2021a)) | $\mathcal{O}\left(\frac{L\Delta_0}{\varepsilon^2}\right)$ | $\mathcal{O}\left(\frac{L}{\mu}\log\frac{1}{\varepsilon}\right)$ |
| | Adagrad (Th. 1 from Défossez et al. (2020)) | $\tilde{\mathcal{O}}\left(\frac{dM^2}{\varepsilon^2}\cdot\frac{L\Delta_0}{\varepsilon^2}\right)$ | — |
| | Adam (Sec. 4.3 from Défossez et al. (2020)) | $\tilde{\mathcal{O}}\left(\frac{dM^2}{\varepsilon^2}\cdot\frac{L\Delta_0}{\varepsilon^2}\right)$ | — |
| | OASIS (Ths. 4.17 and 4.18 from Jahani et al. (2021b)) | $\mathcal{O}\left(\frac{dL^2}{\alpha^2}\cdot\frac{L\Delta_0}{\varepsilon^2}\right)$ | $\mathcal{O}\left(\frac{dL^3}{\alpha^2\mu}\cdot\frac{L}{\mu}\log\frac{1}{\varepsilon}\right)$ [1] |
| | Scaled SARAH with Hutchinson's preconditioner (ours) | $\mathcal{O}\left(\frac{\sqrt{d}L}{\alpha}\cdot\frac{L\Delta_0}{\varepsilon^2}\right)$ | $\mathcal{O}\left(\frac{\sqrt{d}L}{\alpha}\cdot\frac{L}{\mu}\log\frac{1}{\varepsilon}\right)$ |
| | Scaled SARAH with Adam preconditioner (ours) | $\mathcal{O}\left(\frac{M}{\alpha}\cdot\frac{L\Delta_0}{\varepsilon^2}\right)$ | $\mathcal{O}\left(\frac{M}{\alpha}\cdot\frac{L}{\mu}\log\frac{1}{\varepsilon}\right)$ |

| | | Method and reference | Non-convex | Polyak-Łojasiewicz |
|---|---|---|---|---|
| **Stochastic** | **Bounded variance** | SGD (Secs. B.2 and C.2 from Li & Richtárik (2020)) | $\mathcal{O}\left(\frac{L\Delta_0}{\varepsilon^2}+\frac{\sigma^2}{\varepsilon^2}\cdot\frac{L\Delta_0}{\varepsilon^2}\right)$ | $\mathcal{O}\left(\frac{L}{\mu}\log\frac{1}{\varepsilon}+\frac{L\sigma^2}{\mu^2\varepsilon}\right)$ |
| | | Adagrad (Th. 1 from Défossez et al. (2020)) | $\tilde{\mathcal{O}}\left(\frac{dM^2}{\varepsilon^2}\cdot\frac{L\Delta_0}{\varepsilon^2}+d\cdot\frac{\sigma^2}{\varepsilon^2}\cdot\frac{L\Delta_0}{\varepsilon^2}\right)$ | — |
| | | Adam (Sec. 4.3 from Défossez et al. (2020)) | $\tilde{\mathcal{O}}\left(\frac{dM^2}{\varepsilon^2}\cdot\frac{L\Delta_0}{\varepsilon^2}+d\cdot\frac{\sigma^2}{\varepsilon^2}\cdot\frac{L\Delta_0}{\varepsilon^2}\right)$ | — |
| | | OASIS (Ths. 4.17 and 4.18 from Jahani et al. (2021b)) | $\mathcal{O}\left(\frac{dL^2}{\alpha^2}\cdot\frac{L\Delta_0}{\varepsilon^2}+\frac{dL^2}{\alpha^2}\cdot\frac{\sigma^2}{\varepsilon^2}\cdot\frac{L\Delta_0}{\varepsilon^2}\right)$ | $\mathcal{O}\left(\frac{dL^3}{\alpha^2\mu}\cdot\frac{L}{\mu}\log\frac{1}{\varepsilon}+\frac{dL^2}{\alpha^2}\cdot\frac{L\sigma^2}{\mu^2\varepsilon}\right)$ [1] |
| | **Finite-sum** | SARAH (Sec. 3.2 from Pham et al. (2020), Sec. 5 from Li et al. (2021a)) | $\mathcal{O}\left(n+\sqrt{n}\cdot\frac{L\Delta_0}{\varepsilon^2}\right)$ | $\mathcal{O}\left(\left[n+\sqrt{n}\cdot\frac{L}{\mu}\right]\log\frac{1}{\varepsilon}\right)$ |
| | | Scaled SARAH with Hutchinson's preconditioner (ours) | $\mathcal{O}\left(\frac{\sqrt{d}L}{\alpha}\cdot\sqrt{n}\cdot\frac{L\Delta_0}{\varepsilon^2}\right)$ | $\mathcal{O}\left(\left[n+\frac{\sqrt{d}L}{\alpha}\cdot\sqrt{n}\cdot\frac{L}{\mu}\right]\log\frac{1}{\varepsilon}\right)$ |
| | | Scaled SARAH with Adam preconditioner (ours) | $\mathcal{O}\left(\frac{M}{\alpha}\cdot\sqrt{n}\cdot\frac{L\Delta_0}{\varepsilon^2}\right)$ | $\mathcal{O}\left(\left[n+\frac{M}{\alpha}\cdot\sqrt{n}\cdot\frac{L}{\mu}\right]\log\frac{1}{\varepsilon}\right)$ |

[1] for strongly convex problems.

To sum up, our results exceed the estimates already given in the literature for scaled methods. If we take into account that algorithms with preconditioning are strongly attractive from the point of view of real-world learning problems, it turns out that we prove the best results for the practical class of methods at present. Meanwhile, our estimates are still worse than those for unscaled methods, in Appendix G.1 we give some reasoning why these estimates are unimproved. Then we present Section 5 with experiments in which it becomes clear that real problems are not necessarily "the worst", on the contrary, it is on practical problems that our method from Sections 2 and 3 shows itself most strongly.

## 5 NUMERICAL EXPERIMENTS

The purpose of these numerical experiments is to study the practical performance of our new Scaled SARAH and Scaled L-SVRG algorithms, and hence, to understand the advantages of using the proposed diagonal preconditioner on SARAH and L-SVRG. These results will also be compared with SGD, both with and without the preconditioner described in Section 2, as well as the state-of-the-art (first order) preconditioned optimizer Adam.

We test these algorithms on problem (1) with two loss functions: (1) *logistic regression* loss function, which is convex, and (2) *non-linear least squares* loss function, which is nonconvex. The loss functions are described in details below. For further details and experimental results that support the findings of this section, please see Appendix B. Note that all the experiments were initialized at the point $w_0 = 0$, and each experiment was run for 10 different random seeds.

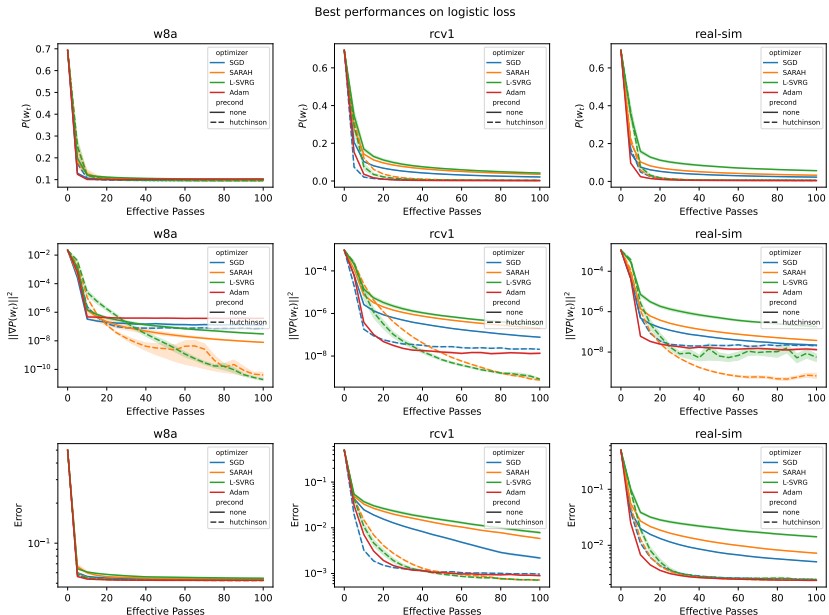

Figure 1: Best performances of the optimizers, including `Adam`, on the (unscaled) LibSVM datasets using the logistic loss. The `Scaled` variants are shown as dashed lines sharing the same color.

## 5.1 LOSS FUNCTIONS

Let $P(w)$ be the empirical risk on a dataset $\{(x_i, y_i)\}_{i=1}^n$ where $x_i \in \mathbb{R}^d$ and $y_i \in \{-1, +1\}$. Then, the *logistic regression loss* is

$$P_{\text{logistic}}(w) = \frac{1}{n}\sum_{i=1}^n \log(1 + e^{-y_i x_i^T w}) \tag{8}$$

whereas for $y_i \in \{0, 1\}$ the *non-linear least squares loss (NLLSQ)* is

$$P_{\text{nllsq}}(w) = \frac{1}{n}\sum_{i=1}^n (y_i - 1/(1 + e^{-x_i^T w}))^2 \tag{9}$$

We consider two different loss functions to test our algorithms on both convex and nonconvex settings.

## 5.2 BINARY CLASSIFICATION ON LIBSVM DATASETS

We train the optimizers on three binary classification LibSVM datasets [3], namely `w8a`, `rcv1`, and `real-sim`. We also consider feature-scaled versions of these datasets, where the scaling is done as follows: we choose a minimum exponent $k_{min}$ and a maximum exponent $k_{max}$, and scale the features by values ranging from $10^{k_{min}}$ to $10^{k_{max}}$ in equal steps in the exponent according to the number of features and in random order. The setting $(k_{min}, k_{max}) = (0, 0)$ corresponds to the original, unscaled version of the datasets. We consider combinations of $k_{min} = 0, -3$ and $k_{max} = 0, 3$. This scaling is done to check the robustness and overall effectiveness of the diagonal preconditioner in comparison with `Adam`. Figure 1 shows the results of the first experiment, and presents three types of line plots for each of the datasets of interest where the loss function is the logistic regression loss (8). Figure 2 shows the same for the NLLSQ loss (9). The first row corresponds to the loss, the second is the squared norm of the gradient, and the third is the error. Tuning was performed in order to select the best hyperparameters (that minimize either the loss, gradient norm squared, or the error). The hyperparameter search grid is reported in Appendix, Table 4, and a thorough discussion is below. We fixed the batch size to be $128$, in order to narrow the fine-tuned variables down to $\eta$, $\beta$, and $\alpha$. Figure 1 shows the performances when minimizing the error on the unscaled datasets, $(k_{min}, k_{max}) = (0, 0)$. Experiments on scaled datasets can be found in Appendix B.2.

Consider the first column in Figure 1, which uses the `w8a` dataset. While preconditioned `SGD` performs better than `SGD`, for `SARAH` and `L-SVRG` this is not the case. Notice that `Adam` performs well initially, but then further effective data passes lead to small improvement (for all three metrics).

---

[3]https://www.csie.ntu.edu.tw/ cjlin/libsvmtools/datasets/

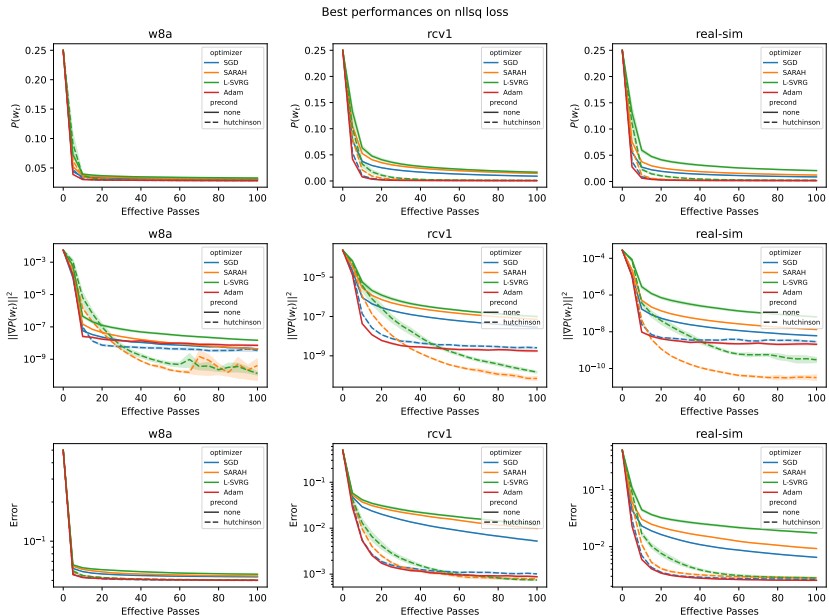

Figure 2: Best performances on the unscaled LibSVM datasets using the NLLSQ loss.

While `SARAH` and `L-SVRG` perform the best on this dataset, after approximately 35 passes for `Scaled L-SVRG` and 60 passes for `Scaled SARAH`, performance is better than `Adam`. For the remaining three datasets, preconditioning helps in all cases, with `Scaled L-SVRG` and `Scaled SARAH`, performing the best.

In order to understand the main factors that affect the preconditioner, we ran comparative studies, including studying the parameters $\beta$, and $\alpha$, as well as studying the initialization of preconditioner $D_0$, including a warm-up period, and studying how the number of samples, $z$, impacts performance. First, recalling (6), there did not appear to be any significant improvement from averaging across more samples of $z$ per minibatch, neither in the initialization nor in the update step. We also observed that initializing $D_0$ with a batch size of 100 was sufficiently good for non-sparse problems, and consistently resulted in a relative error of within 0.1 from the true diagonal. However, increasing the number of warm-up samples, proportionally to the number of features, led to observable improvements in convergence for sparse datasets.

We also investigated the role that $\beta$ (3) played in algorithm performance. We found that larger values lead to slightly slower but more stable convergence. The best $\beta$ highly depends on the dataset, but the value 0.999 appeared to be a good starting point in general. To ensure a fair comparison, we also optimized `Adam`'s momentum parameter $\beta_2$ over the same range.

Aside from the batch size and learning rate, we found that, for ill-conditioned problems, the choice of $\alpha$ (recall (5) and Remark 2.1) played an important role in determining the quality of the solution, convergence speed, and stability (which is not obvious from Figure 1). For example, if the features were scaled with $k_{min} = -3$ and $k_{max} = 0$, the best $\alpha$ is often around $10^{-7}$ (very small), whereas if we scaled with $k_{min} = 0$ and $k_{max} = 3$, the best $\alpha$ becomes $10^{-1}$ (relatively large). Therefore, finding the best $\alpha$ might require some additional fine-tuning, depending on the choice of $\eta$ and $\beta$. However, we noticed that once we had tuned the learning rate for one scaled version of the dataset, the same learning rate transferred well to all the other scaled versions. In general, the optimal learning rate in our `Scaled` algorithms is very robust to feature scaling, given that $\alpha$ is chosen well, whereas `Adam`'s learning rate depends more heavily upon how ill-conditioned the problem is, so it requires fine-tuning across a potentially much wider range. In our case, tuning $\alpha$ and $\beta$ is straightforward, so we obtained state-of-the-art performance with minimal parameter tuning.

The choice $\beta = 1 - 1/(t + 1)$ was investigated in Appendices B.5 and G.3. Preliminary results suggest that this choice virtually removes the dependence on $\alpha$, and is competitive with a fine-tuned $\beta$ across a large number of values. This makes fine-tuning much easier, even for strong feature scaling.

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

# A   Scaled L-SVRG

SVRG (Johnson & Zhang, 2013; Xiao & Zhang, 2014) is a variance reduced stochastic gradient method that is very popular for finite sum optimization problems. However, the algorithm has a double loop structure, and careful tuning of hyper-parameters is required for good practical performance.

Recently, Hofmann et al. (2015) proposed a Loopless SVRG (L-SVRG) variant, that has a simpler, single loop structure, which can be applied to problem (1) in the convex and smooth case. This was extended in Qian et al. (2021) to cover the composite case with an arbitrary sampling scheme. With its single loop structure, and consequently fewer hyperparameters to tune, coupled with the fact that, unlike for PAGE (recall Section 1), L-SVRG uses an *unbiased* estimate of the gradient, L-SVRG is a versatile and competitive algorithm for problems of the form (1).

However, as for the other previously mentioned gradient based methods, L-SVRG can perform poorly when the problem is badly scaled and/or ill-conditioned. This provides the motivation for the Scaled L-SVRG method that we propose in this work. Our Scaled L-SVRG algorithm combines the positive features of L-SVRG, with a preconditioner, to give a method that is loopless, has few hyperparameters to tune, uses an unbiased estimate of the gradient, and adaptively scales the search direction depending upon the local curvature. The Scaled L-SVRG method is presented now as Algorithm 2.

---

**Algorithm 2** Scaled L-SVRG

---

1: **Input:** initial point $w_0$, learning rate $\eta$, preconditioner $\hat{D}_0$, probability $p$
2: $z_0 = w_0$, $v_0 = \nabla P(w_0)$
3: **for** $t = 0, 1, 2, \ldots$ **do**
4: $\quad w_{t+1} = w_t - \eta \hat{D}_t^{-1} v_t$
5: $\quad z_{t+1} = \begin{cases} z_t, & \text{with probability } p \\ w_t, & \text{with probability } 1 - p \end{cases}$
6: $\quad$ Generate independently batches $i_{t+1}$ for $v_{t+1}$ and $\mathcal{J}_t$ for $\hat{D}_{t+1}$
7: $\quad v_{t+1} = \nabla f_{i_{t+1}}(w_{t+1}) - \nabla f_{i_{t+1}}(z_{t+1}) + \nabla P(z_{t+1})$
8: $\quad$ Update the preconditioner $\hat{D}_{t+1}$
9: **end for**
10: **Output:** $\hat{w}_T$ chosen uniformly from $\{w_t\}_{t=0}^T$

---

Scaled L-SVRG can be described, in words, as follows. The algorithm is initialized with an initial point $w_0$, a learning rate $\eta$, an initial preconditioner $\hat{D}_0$, and probability $p$. At each iteration $t \geq 0$ of Scaled L-SVRG (Algorithm 2) a search direction $v_t$ is generated. This is made up of the full gradient plus a small adjustment. Next, the new point $w_{t+1}$ is taken to be a step from $w_t$ in the *scaled* direction $\hat{D}_t^{-1} v_t$, of size $\eta$. The new point $z_{t+1}$ is either the (unchanged) previous point $z_t$ with probability $p$, or the scaled approximate gradient step $w_{t+1}$ with probability $1 - p$. Finally, the preconditioner is updated and the next iterate begins. The output, denoted by $\hat{w}_T$ is chosen uniformly from the points $w_t$, for $t = 0, \ldots, T$, generated by Scaled L-SVRG (Algorithm 2).

Note that a key difference between L-SVRG (Hofmann et al., 2015; Qian et al., 2021) and our new Scaled L-SVRG is the inclusion of the preconditioner in Step 4; recall that a competitive preconditioner is described in Section 2.

The following theorem presents a complexity bound on the number of iterations required by Scaled L-SVRG to obtain an $\varepsilon$-optimal solution for the non-convex problem (1).

**Theorem A.1.** *Suppose that Assumption 1.1 holds, let $\varepsilon > 0$, let $p$ denote the probability and let the step-size satisfy $\eta \leq \min\left\{ \frac{\alpha}{4L}, \frac{\sqrt{p}\alpha}{\sqrt{24}L}, \frac{p^{2/3}}{144^{2/3}} \frac{\alpha}{L} \right\}$. Given an initial point $w_0 \in \mathbb{R}^d$, let $\Delta_0 = P(w_0) - P^*$. Then the number of iterations performed by* Scaled L-SVRG, *starting from $w_0$, required to obtain an $\varepsilon$-approximate solution of non-convex finite-sum problem (1) can be bounded by*

$$T = \mathcal{O}\left( \frac{\Gamma}{\alpha} \frac{L\Delta_0}{p^{2/3}\varepsilon^2} \right).$$

While the previous theorem held under a smoothness assumption, here we prove a complexity result for Scaled SARAH under both smoothness *and* PL assumptions.

**Theorem A.2.** *Suppose that Assumptions 1.1 and 1.2 hold, let $\varepsilon > 0$, let $p$ denote the probability and let the step-size satisfy $\eta \leq \min \left\{ \frac{p\Gamma}{6\mu}, \frac{1}{4}\frac{\alpha}{L}, \left(\frac{p}{6}\right)^{1/2}\frac{\alpha}{L}, \left(\frac{p}{6}\right)^{2/3}\frac{\alpha}{L} \right\}$. Then the number of iterations performed by* `Scaled L-SVRG` *sufficient for finding an $\varepsilon$-approximate solution of non-convex finite-sum problem* (1) *can be bounded by*

$$T = \mathcal{O}\left( \max\left\{ \frac{1}{p}, \frac{\Gamma}{\alpha}\frac{L}{p^{2/3}\mu} \right\} \log \frac{\Delta_0}{\varepsilon} \right).$$

We give corollaries on the stochastic gradient of complexities.

**Corollary A.3.** *Suppose that Assumption 1.1 holds, let $\varepsilon > 0$, let $p = \frac{1}{n+1}$ and let the step-size satisfy $\eta \leq \min \left\{ \frac{\alpha}{4L}, \frac{\sqrt{p}\alpha}{\sqrt{24}L}, \frac{p^{2/3}}{144^{2/3}}\frac{\alpha}{L} \right\}$. Given an initial point $w_0 \in \mathbb{R}^d$, let $\Delta_0 = P(w_0) - P^*$. Then the stochastic gradient complexity performed by* `Scaled L-SVRG`, *starting from $w_0$, required to obtain an $\varepsilon$-approximate solution of non-convex finite-sum problem* (1) *can be bounded by $\mathcal{O}\left( n + \frac{\Gamma}{\alpha}\frac{L\Delta_0}{\varepsilon^2}n^{2/3} \right)$.*

**Corollary A.4.** *Suppose that Assumptions 1.1 and 1.2 hold, let $\varepsilon > 0$, let $p = \frac{1}{n+1}$ and let the step-size satisfy $\eta \leq \min \left\{ \frac{p\Gamma}{6\mu}, \frac{1}{4}\frac{\alpha}{L}, \left(\frac{p}{6}\right)^{1/2}\frac{\alpha}{L}, \left(\frac{p}{6}\right)^{2/3}\frac{\alpha}{L} \right\}$. Then the stochastic gradient complexity performed by* `Scaled L-SVRG` *sufficient for finding an $\varepsilon$-approximate solution of non-convex finite-sum problem* (1) *can be bounded by $\mathcal{O}\left( \left\{ n + \frac{\Gamma}{\alpha}\frac{L}{\mu}n^{2/3} \right\} \log \frac{\Delta_0}{\varepsilon} \right)$.*

## B  ADDITIONAL NUMERICAL EXPERIMENTS

Here we provide additional details related to our experimental set-up, as well as presenting the results of additional numerical experiments.

### B.1  BEST PERFORMANCE GIVEN A FIXED PARAMETER

Table 4 states the hyperparameters that were used in our numerical experiments.

Table 4: Hyperparameter search grid.

| **Settings** | |
| --- | --- |
| Optimizer | `SARAH, L-SVRG, SGD, Adam` |
| `Scaled` | `True, False` (except for `Adam`) |
| Dataset | `a9a, w8a, rcv1, real-sim` |
| $(k_{min}, k_{max})$ | $(0, 0), (0, 3), (-3, 0), (-3, 3)$ |
| Loss function | `logistic, NLLSQ` |
| Random Seed | $0, \cdots, 9$ |
| **Parameters** | |
| Batch Size | $128, 512$ |
| $\eta$ | $2^{-20}, 2^{-18}, \cdots, 2^4$ |
| $\alpha$ | $10^{-1}, 10^{-3}, 10^{-7}$ |
| $\beta$ | $0.95, 0.99, 0.995, 0.999, $ `avg` |

We ran extensive experiments in order to find the best performing set of parameters for each optimizer on each dataset. For our parameter search, we fixed one parameter (e.g., $\alpha = 10^{-1}$), and then found the best choice for the remaining parameters, given that fixed value. This allowed us to understand how robust the algorithm's optimal performance is, with respect to each parameter. In other words, 'how does changing one parameter degrade the quality of the solution or the overall performance of the algorithm?'. We also report the best overall performances.

Here, we consider fixing one of three parameters: $\eta$, $\beta$, and $\alpha$, and then plot the trajectory of the gradient norm squared $\|\nabla P(w_t)\|^2$ for the setting that minimizes the error with respect to the other parameters. The values '$\alpha$ = none' and '$\beta$ = none' indicate non-preconditioned trajectories, and the value $\beta$ = avg indicate the choice $\beta_t = 1 - 1/(t+1)$ (or more precisely, as described in Appendix B.5). See Figures 3, 4, 5, and 6.

## B.2 CORRUPTING SCALE OF FEATURES

We consider the settings where the features of the data are corrupted with a given logarithmic scale. We show the best overall performances on 4 scaled datasets with $(k_{min}, k_{max}) \in \{(-3, 0), (0, 3), (-3, 3)\}$. This allowed us to understand the impact of no preconditioning versus preconditioning on poorly scaled problems. The results are shown in Figures 7, 8, and 9.

## B.3 CONVEX VS. NON-CONVEX LOSS

We test our algorithms on the non-linear least squares loss, which is a non-convex loss function. We show the best performance on the unscaled datasets, as well as datasets scaled with $(k_{min}, k_{max}) = (-3, 3)$. See Figures 10, and 11.

## B.4 RELATIVE ERROR OF $D_0$

For the preconditioner described in Section 2, the initial preconditioner $D_0$ must be chosen appropriately. Note that, if $\texttt{diag}(H_0)$ is the true Hessian diagonal for some initial point $w_0$, then the relative error of the approximation $D_0$ to $\texttt{diag}(H_0)$ is

$$\frac{\|D_0 - \texttt{diag}(H_0)\|}{\|\texttt{diag}(H_0)\|}.$$

For our numerical experiments we used this relative error to determine whether the initial preconditioner $D_0$ was sufficiently accurate. In particular, we noticed that, if a minibatch of size 100 was used, then the resulting $D_0$ almost always reached a relative error of below 0.1, for dense datasets. For sparse datasets, more warm-up samples can improve convergence and quality of the solution, but in either case, a minibatch of size 100 was sufficient to establish convergence. Thus, we initialized $D_0$ with a size 100 minibatch in most of our experiments. See Figures 12, 13, and 14.

## B.5 CHOOSING $\beta_t$

We show plots where we finetune $\alpha$ and $\beta$ on a wider range of parameters. Similarly to Section B.1, we optimize the other parameters by fixing the parameter of interest at the chosen value, and report the optimal performance metric. We only consider `Scaled SARAH` in this experiment. We show plots where we minimize either the error or the gradient norm under the logistic loss or NLLSQ loss. We also show the optimal performance when choosing $\beta_t = 1 - \frac{1}{t+t_0}$, where $t_0$ is the number of warm-up samples. We run experiments for 10 random seeds and show the confidence intervals for each hyperparameter setting. We see that the suggested choice for $\beta_t$ is consistently competitive with the best $\beta$ and is robust to feature scaling as well. See Figures 15, 16, and 17.

## B.6 DEEP NEURAL NETS ON MNIST

In order to demonstrate that our algorithms are indeed practical and competitive with the state-of-the-art, we test them on deep neural nets. The setting of our experiment is widely accessible and well-known; train the LeNet-5 model on MNIST dataset with the cross entropy loss Lecun et al. (1998). For this experiment, we believe that it suffices to test `Scaled L-SVRG` vs. `Adam`. For `L-SVRG`, we use $p = 0.999$. The programming framework we run this experiment on is PyTorch Paszke et al. (2019). The hyperparameter search grid is slightly reduced for this experiment. To be specific, learning rates bigger than $2^{-6}$ and lower than $2^{-14}$ are omitted, and we only consider $\beta$ in $\{\text{avg}, 0.99, 0.999\}$. The result is shown in Figure 18 and 19. We observe that the performance of `Scaled L-SVRG` is indeed competitive with `Adam` in this simple deep learning problem. We omit trajectories that diverge in this plot, which is why the value $\alpha = 10^{-7}$ is not seen in Figure 19. In the future, we plan on running more sophisticated deep learning experiments, and explore ways to adapt our algorithms to these settings in particular.

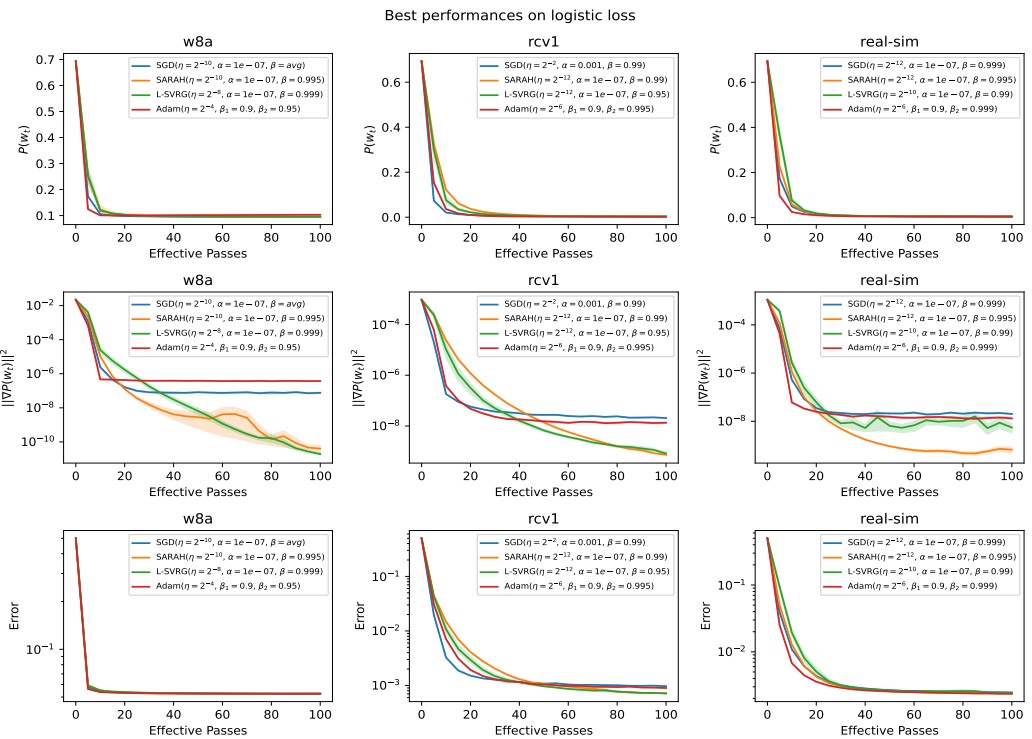

Figure 3: Performance of the best parameters minimizing the error using the logistic loss.

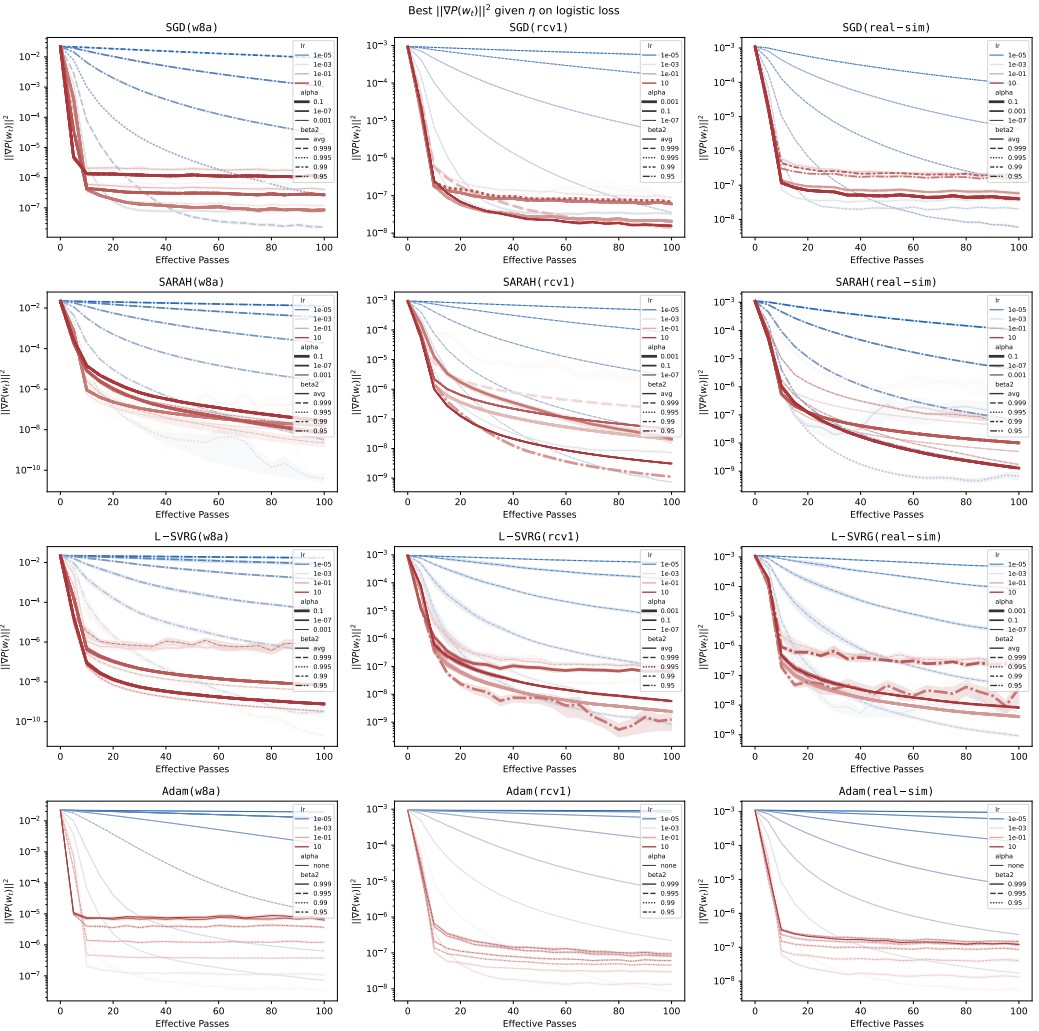

Figure 4: Trajectory of $\|\nabla P(w_t)\|^2$ of the best parameters minimizing the error given $\eta$ and logistic loss.

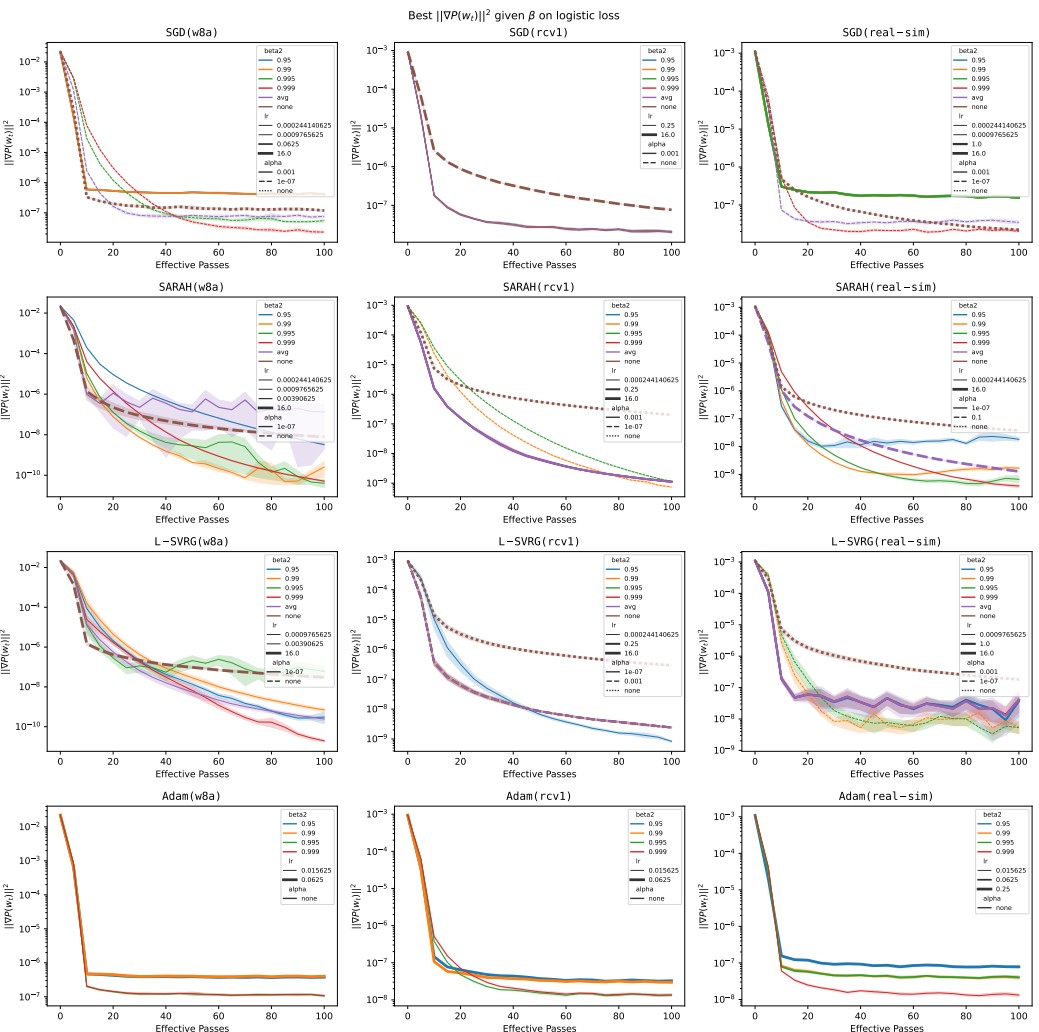

Figure 5: Trajectory of $\|\nabla P(w_t)\|^2$ of the best parameters minimizing the errors given $\beta$ and logistic loss.

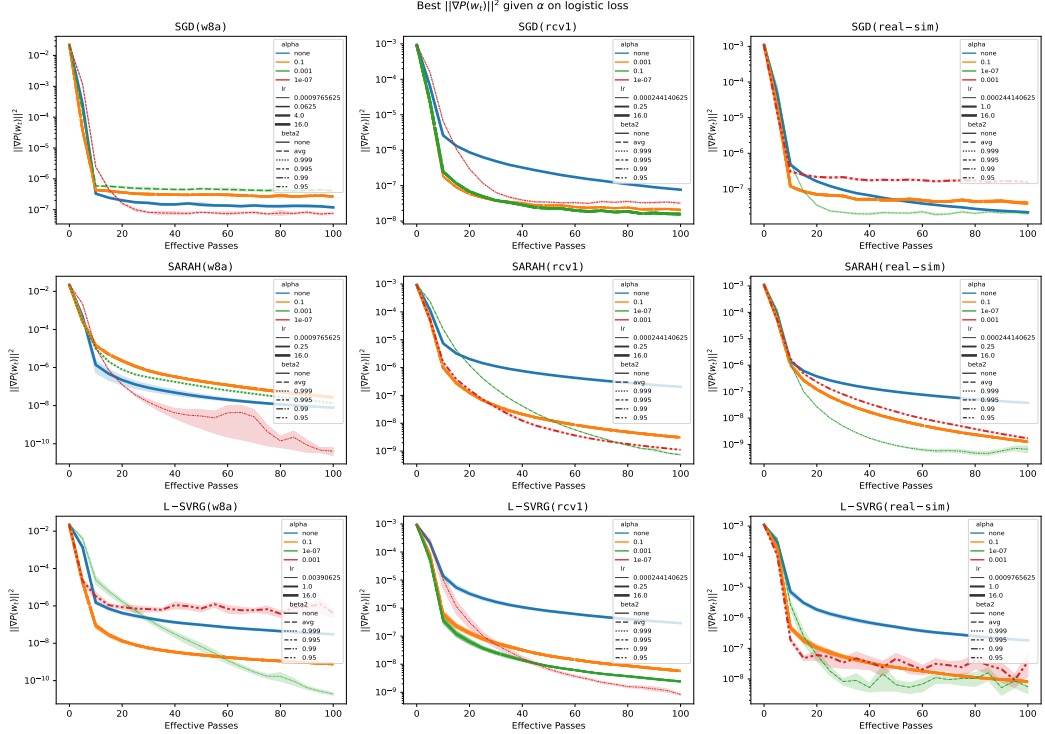

Figure 6: Trajectory of $\|\nabla P(w_t)\|^2$ of the best parameters minimizing the error given $\alpha$ and logistic loss.

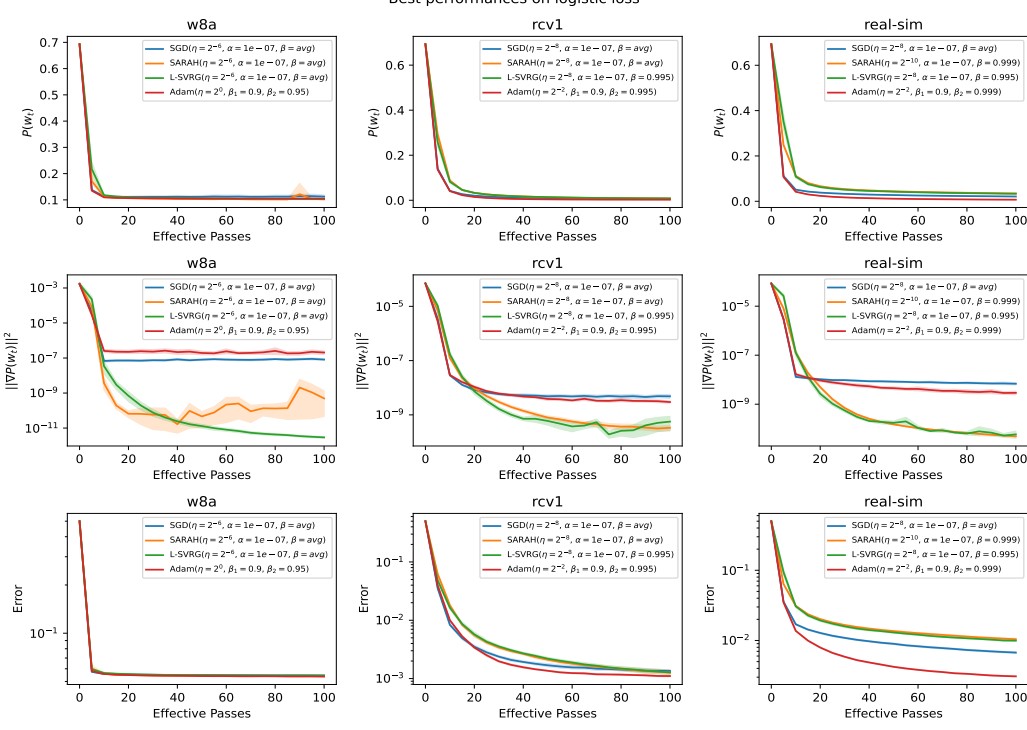

Figure 7: Performance of the best parameters minimizing the error given $(k_{min}, k_{max}) = (-3, 0)$.

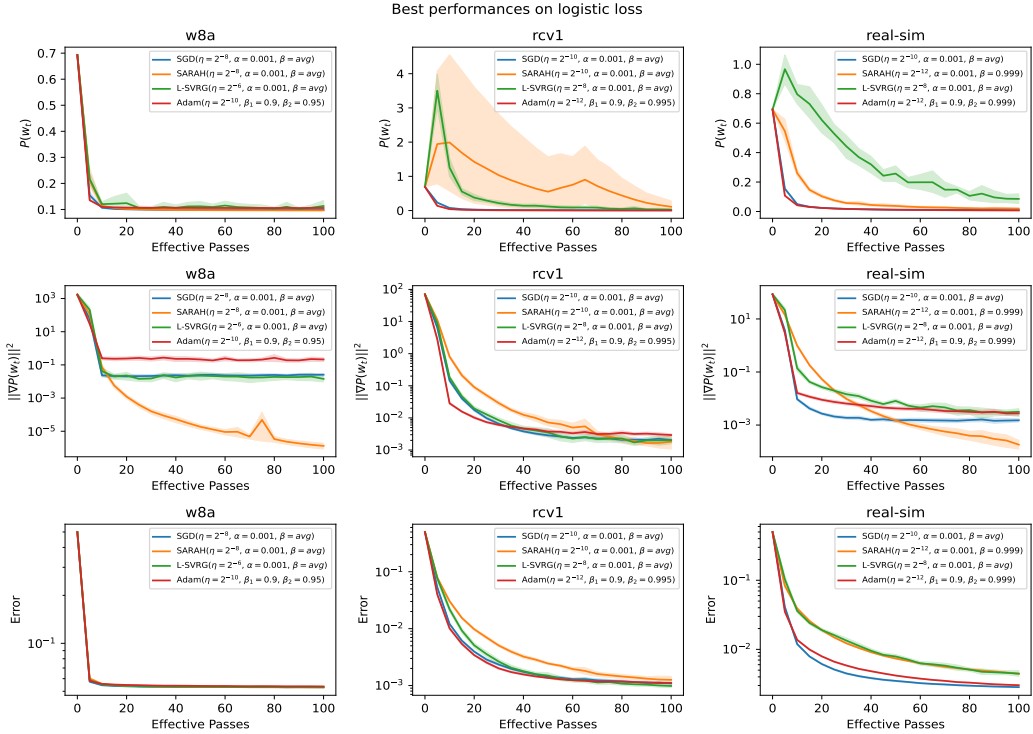

Figure 8: Performance of the best parameters minimizing the error given $(k_{min}, k_{max}) = (0, 3)$.

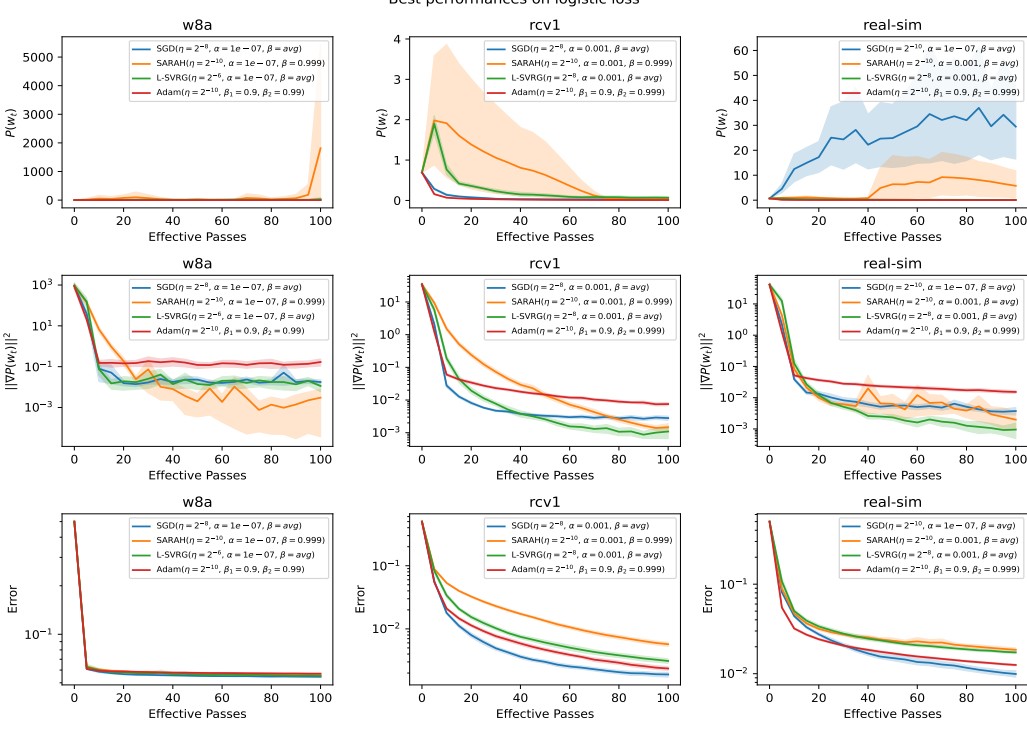

Figure 9: Performance of the best parameters minimizing the error given $(k_{min}, k_{max}) = (-3, 3)$.

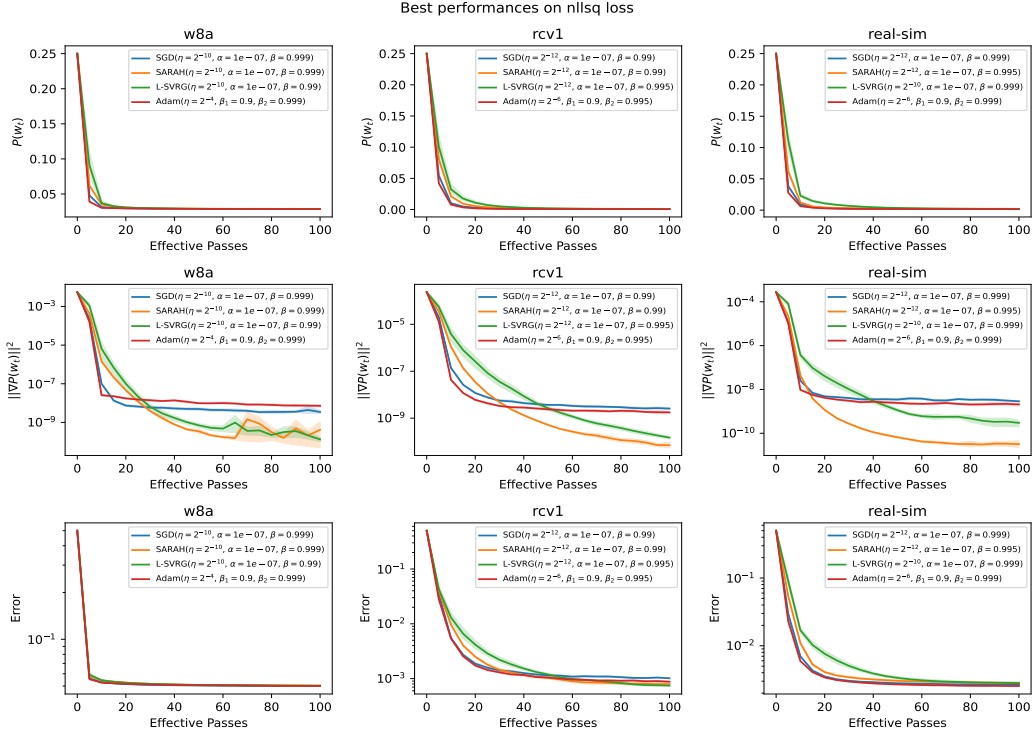

Figure 10: Performance of the best parameters minimizing the error using the NLLSQ loss.

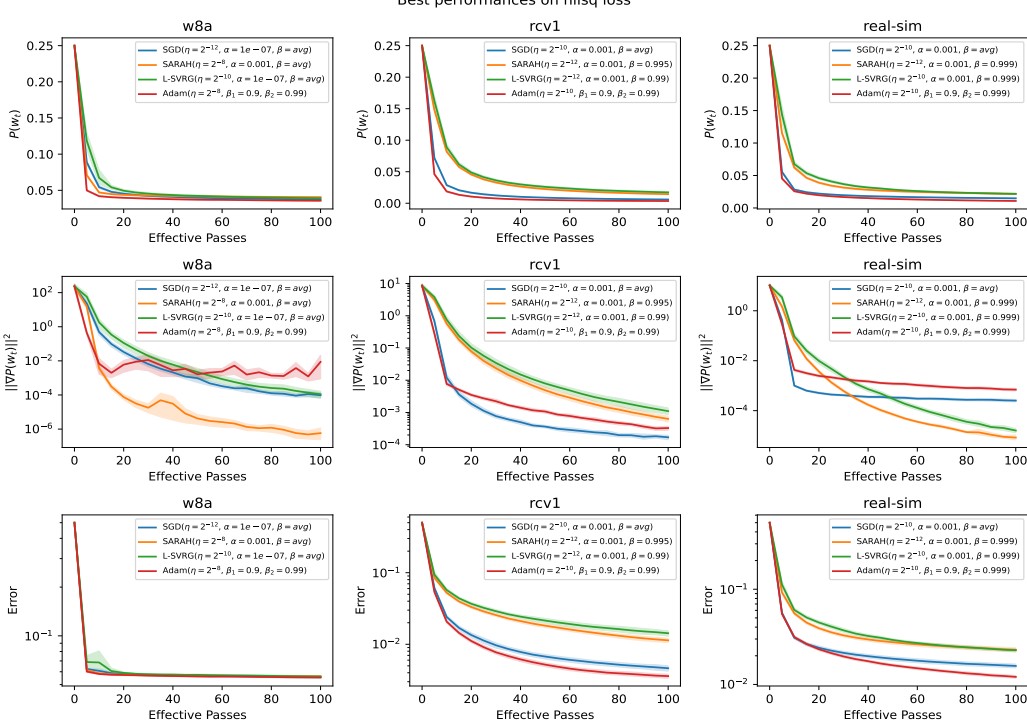

Figure 11: Performance of the best parameters minimizing the error on NLLSQ loss given $(k_{min}, k_{max}) = (-3, 3)$.

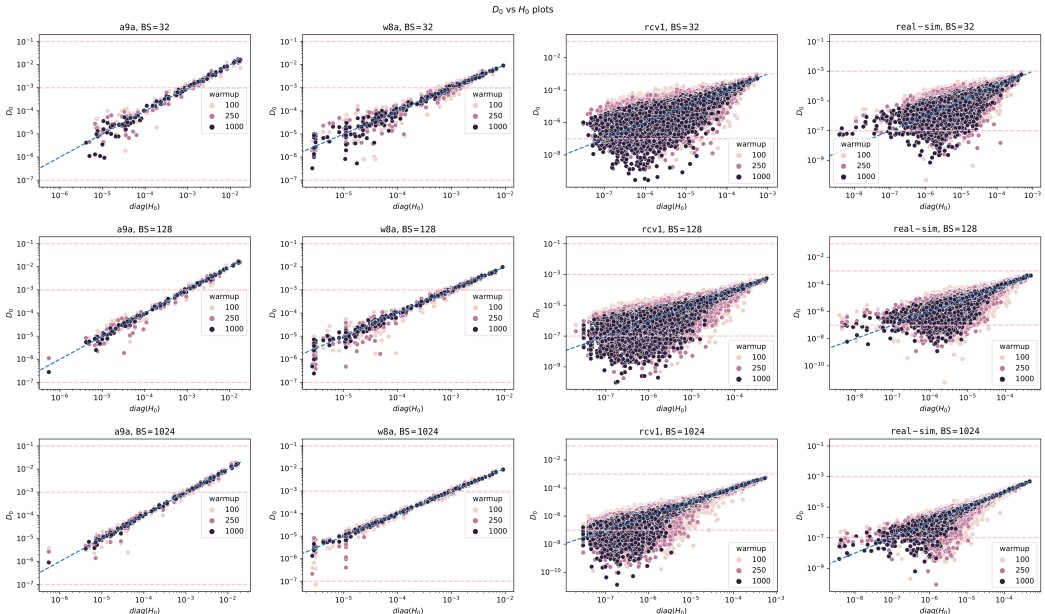

Figure 12: The diagonal values of the true hessian $H_0$ vs. the estimate $D_0$. The tree $\alpha$ levels in our hyperparameter search grid are shown in a dashed pink horizonatal lines. Increasing the number of warmup samples is beneficial, with slightly diminishing benefits as the batch size decreases. This can be seen from the larger number of light pink points around the diagonal for batch size 1024 on `real-sim`.

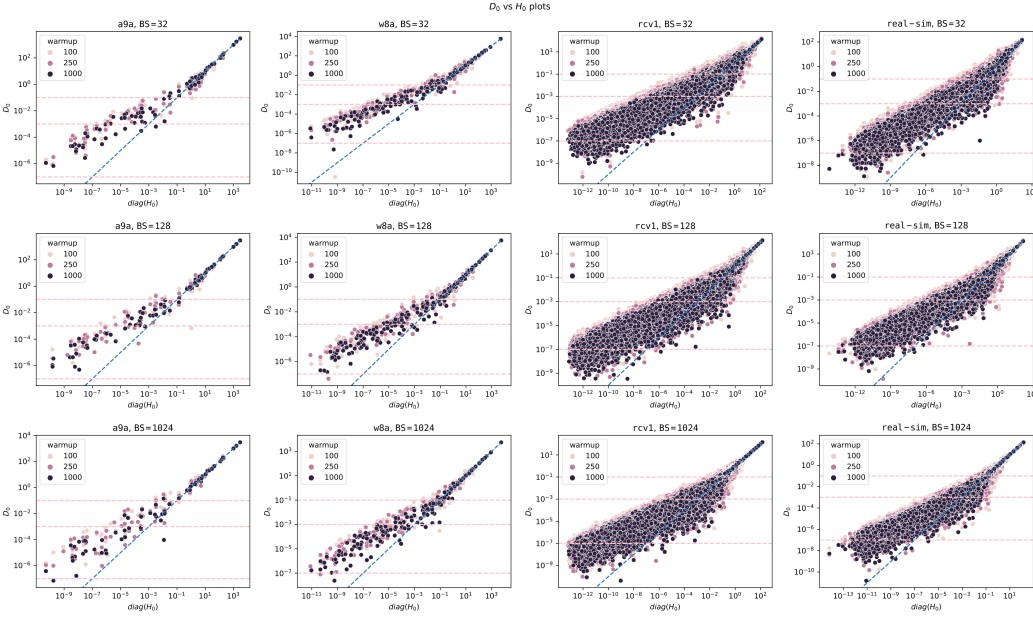

Figure 13: The diagonal values of the true hessian $H_0$ vs. the estimate $D_0$ given $(k_{min}, k_{max}) = (-3, 3)$.

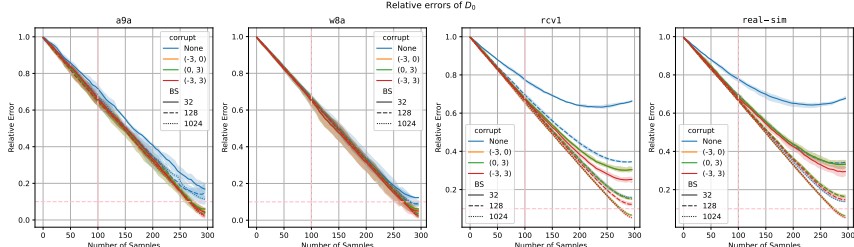

Figure 14: The relative error of the diagonal with respect to the diagonal of the true Hessian plotted against the number of warmup samples. The value `corrupt` indicates the feature scaling. For sparse datasets, it is more difficult to decrease the relative error below $0.1$.

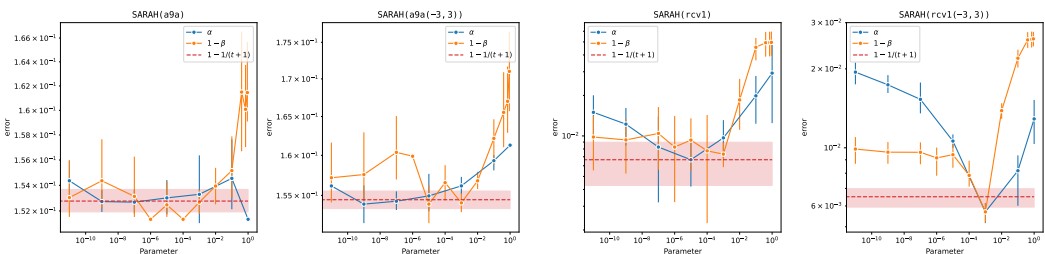

Figure 15: Optimal error achieved with the logistic loss given the value of the parameter. The optimal error for the "averaging" $\beta_t$ is shown as a dashed red line. The arguments after the dataset name are the scaling factors.

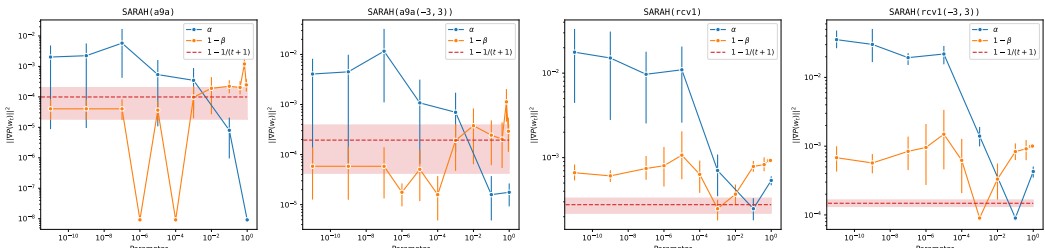

Figure 16: Optimal gradient norm achieved with the logistic loss given the value of the parameter. The optimal gradient norm for the "averaging" $\beta_t$ is shown as a dashed red line. The arguments after the dataset name are the scaling factors.

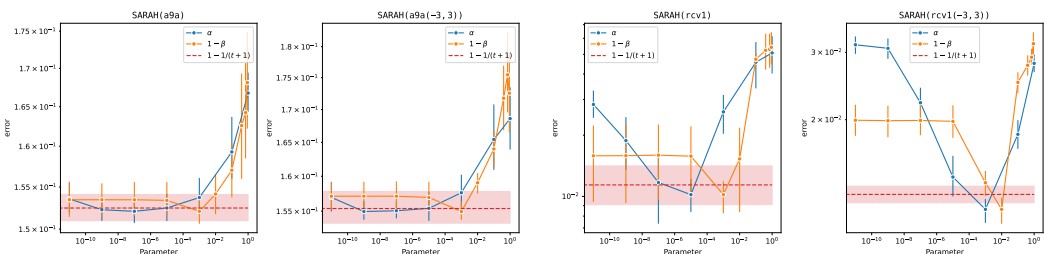

Figure 17: Optimal error achieved with the NLLSQ loss given the value of the parameter. The optimal error for the "averaging" $\beta_t$ is shown as a dashed red line. The arguments after the dataset name are the scaling factors.

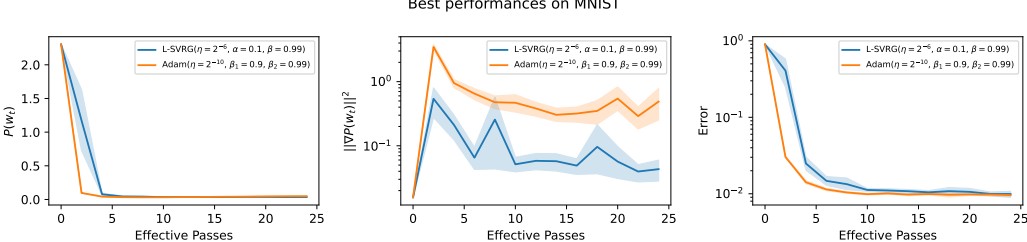

Figure 18: Best performance of `Scaled L-SVRG` and `Adam` on MNIST.

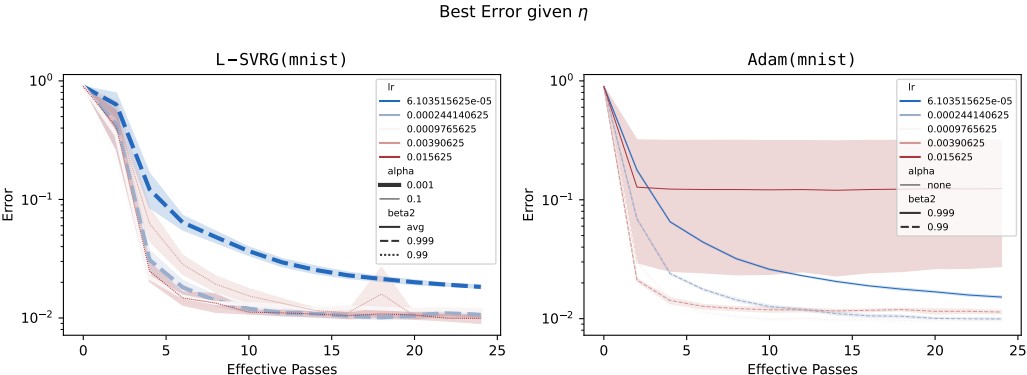

Figure 19: Best error given $\eta$ on MNIST.

## C  LEMMA 2.1

In this section we prove Lemma 2.1.

**Proof:** To begin, note that by constructing matrices $\hat{D}_t$ according to (3) – (6), $\hat{D}_t$ is a diagonal matrix with positive elements on the diagonal, where every elements is at least $\alpha$. Hence, $\alpha I \preccurlyeq \hat{D}_t$.

To prove that $\hat{D}_t \preccurlyeq \Gamma I$, let us show that all elements of $\hat{D}_t$ do not exceed $\Gamma$. Based on (3), it suffices to show that $\|\text{diag}\left(z_t \odot \nabla^2 P_{\mathcal{J}_t}(w_t)z_t\right)\|_\infty \leq \Gamma$ for all $t$.

$$\|\text{diag}\left(z_t \odot \nabla^2 P_{\mathcal{J}_t}(w_t)z_t\right)\|_\infty^2 \leq \|\left(z_t \odot \nabla^2 P_{\mathcal{J}_t}(w_t)z_t\right)\|_\infty^2$$

$$\leq \left(\max_i \left[\sum_{j=1}^d |(\nabla^2 P_{\mathcal{J}_t}(w_t))_{ij}|\right]\right)^2$$

$$\leq \max_i \left(\sum_{j=1}^d |(\nabla^2 P_{\mathcal{J}_t}(w_t))_{ij}|\right)^2$$

$$\leq \max_i \left[d \sum_{j=1}^d (\nabla^2 P_{\mathcal{J}_t}(w_t))_{ij}^2\right]$$

$$\leq d \sum_{i=1}^d \sum_{j=1}^d (\nabla^2 P_{\mathcal{J}_t}(w_t))_{ij}^2$$

$$\leq d\|\nabla^2 P_{\mathcal{J}_t}(w_t)\|_2^2 \leq dL^2.$$

In the last step we use $L$-smoothness of $f_i$ (Assumption 1.1). Finally, we have $\|\text{diag}\left(z_t \odot \nabla^2 P_{\mathcal{J}_t}(w_t)z_t\right)\|_\infty \leq \sqrt{d}L = \Gamma$.

$\square$

## D  LEMMA 2.1 FOR ADAM

In this section we prove Lemma 2.1, but for Adam preconditioning. The rules for calculating the $\hat{D}$-matrix for Adam method are as follows:

$$D_t^2 = \beta_t D_{t-1}^2 + (1-\beta_t)H_{\mathcal{J}_t}^2, \quad D_0^2 = \frac{1}{|\mathcal{J}_0|} \sum_{j \in \mathcal{J}_0} \text{diag}(\nabla f_j(w_0) \odot \nabla f_j(w_0))$$

where

$$\beta_t = \frac{\beta - \beta^{t+1}}{1 - \beta^{t+1}} \quad \text{with} \quad \beta \in (0;1), \quad H_{\mathcal{J}_t}^2 = \frac{1}{|\mathcal{J}_t|} \sum_{j \in \mathcal{J}_t} \text{diag}(\nabla f_j(w_t) \odot \nabla f_j(w_t)),$$

and

$$\left(\hat{D}_t\right)_{i,i} = \max\{\alpha, |D_t|_{i,i}\}.$$

**Lemma D.1.** *For any $t \geq 1$, we have $\alpha I \preccurlyeq \hat{D}_t \preccurlyeq \Gamma I$, where $0 < \alpha \leq \Gamma = M$.*

**Proof:** To begin, note that by constructing matrices $\hat{D}_t$ according to these rules, $\hat{D}_t$ is a diagonal matrix with positive elements on the diagonal, where every elements is at least $\alpha$. Hence, $\alpha I \preccurlyeq \hat{D}_t$.

Moreover, using the diagonal structure of $H^2_{\mathcal{J}_t}$, one can note that

$$
\begin{aligned}
\|H^2_{\mathcal{J}_t}\|_\infty &= \left\| \frac{1}{|\mathcal{J}_t|} \sum_{j \in \mathcal{J}_t} \text{diag}(\nabla f_j(w_t) \odot \nabla f_j(w_t)) \right\|_\infty \\
&\leq \frac{1}{|\mathcal{J}_t|} \sum_{j \in \mathcal{J}_t} \|\text{diag}(\nabla f_j(w_t) \odot \nabla f_j(w_t))\|_\infty \\
&= \frac{1}{|\mathcal{J}_t|} \sum_{j \in \mathcal{J}_t} \|\nabla f_j(w_t) \odot \nabla f_j(w_t)\|_\infty \\
&= \frac{1}{|\mathcal{J}_t|} \sum_{j \in \mathcal{J}_t} \|\nabla f_j(w_t)\|^2_\infty \\
&\leq \frac{1}{|\mathcal{J}_t|} \sum_{j \in \mathcal{J}_t} \|\nabla f_j(w_t)\|^2_2 \leq M^2.
\end{aligned}
$$

Finally, we have $\|H^2_{\mathcal{J}_t}\|_\infty \leq M^2$. And then, $\alpha I \preccurlyeq \hat{D}_t \preccurlyeq MI = \Gamma I$.

$\square$

# E  Scaled SARAH

In this section we present the proofs for the main theoretical complexity results for the `Scaled SARAH` from Section 3.

**Lemma E.1** (Descent Lemma). *Suppose that function $P$ satisfies Assumption 1.1 and Algorithm 1 generate a sequence $\{w_t\}_{t \geq 0}$. Then we have for any $t \geq 0$ and $\eta$*

$$
P(w_{t+1}) \leq P(w_t) + \frac{\eta}{2\alpha}\|\nabla P(w_t) - v_t\|^2 - \left(\frac{1}{2\eta} - \frac{L}{2\alpha}\right)\|w_{t+1} - w_t\|^2_{\hat{D}_t} - \frac{\eta}{2}\|\nabla P(w_t)\|^2_{\hat{D}_t^{-1}}.
$$

**Proof:** Using $L$-smoothness of the function $P$ and $I \preccurlyeq \frac{1}{\alpha}\hat{D}_t$

$$
\begin{aligned}
P(w_{t+1}) &\leq P(w_t) + \langle \nabla P(w_t), w_{t+1} - w_t \rangle + \frac{L}{2}\|w_{t+1} - w_t\|^2 \\
&\leq P(w_t) + \langle \nabla P(w_t), w_{t+1} - w_t \rangle + \frac{L}{2\alpha}\|w_{t+1} - w_t\|^2_{\hat{D}_t}.
\end{aligned}
$$

With an update of `Scaled SARAH`: $w_{t+1} = w_t - \eta\hat{D}_t^{-1}v_t$, we get

$$
\begin{aligned}
P(w_{t+1}) \leq\ & P(w_t) + \langle \nabla P(w_t) - v_t, -\eta\hat{D}_t^{-1}v_t \rangle + \frac{1}{\eta}\langle \hat{D}_t(w_t - w_{t+1}), w_{t+1} - w_t \rangle \\
& + \frac{L}{2\alpha}\|w_{t+1} - w_t\|^2_{\hat{D}_t} \\
=\ & P(w_t) + \eta\langle \nabla P(w_t) - v_t, \nabla P(w_t) - v_t \rangle_{\hat{D}_t^{-1}} \\
& - \eta\langle \nabla P(w_t) - v_t, \nabla P(w_t) \rangle_{\hat{D}_t^{-1}} - \left(\frac{1}{\eta} - \frac{L}{2\alpha}\right)\|w_{t+1} - w_t\|^2_{\hat{D}_t} \\
=\ & P(w_t) + \eta\|\nabla P(w_t) - v_t\|^2_{\hat{D}_t^{-1}} - \eta\langle \nabla P(w_t) - v_t, \nabla P(w_t) \rangle_{\hat{D}_t^{-1}} \\
& - \left(\frac{1}{\eta} - \frac{L}{2\alpha}\right)\|w_{t+1} - w_t\|^2_{\hat{D}_t}.
\end{aligned}
$$

Let us define $\bar{w}_{t+1} = w_t - \eta \hat{D}_t^{-1} \nabla P(w_t)$. Using this new notation and an update $w_{t+1} = w_t - \eta \hat{D}_t^{-1} v_t$, then we have

$$
\begin{aligned}
P(w_{t+1}) \leq & P(w_t) + \eta \|\nabla P(w_t) - v_t\|_{\hat{D}_t^{-1}}^2 - \frac{1}{\eta} \langle w_{t+1} - \bar{w}_{t+1}, w_t - \bar{w}_{t+1} \rangle_{\hat{D}_t} \\
& - \left( \frac{1}{\eta} - \frac{L}{2\alpha} \right) \|w_{t+1} - w_t\|_{\hat{D}_t}^2 \\
= & P(w_t) + \eta \|\nabla P(w_t) - v_t\|_{\hat{D}_t^{-1}}^2 - \left( \frac{1}{\eta} - \frac{L}{2\alpha} \right) \|w_{t+1} - w_t\|_{\hat{D}_t}^2 \\
& - \frac{1}{2\eta} \left( \|w_{t+1} - \bar{w}_{t+1}\|_{\hat{D}_t}^2 + \|w_t - \bar{w}_{t+1}\|_{\hat{D}_t}^2 - \|w_{t+1} - w_t\|_{\hat{D}_t}^2 \right) \\
= & P(w_t) + \eta \|\nabla P(w_t) - v_t\|_{\hat{D}_t^{-1}}^2 - \left( \frac{1}{\eta} - \frac{L}{2\alpha} \right) \|w_{t+1} - w_t\|_{\hat{D}_t}^2 \\
& - \frac{1}{2\eta} \left( \eta^2 \|\nabla P(w_t) - v_t\|_{\hat{D}_t^{-1}}^2 + \eta^2 \|\nabla P(w_t)\|_{\hat{D}_t^{-1}}^2 - \|w_{t+1} - w_t\|_{\hat{D}_t}^2 \right).
\end{aligned}
$$

$\square$

**Lemma E.2.** *Suppose that the function $P$ satisfies satisfies Assumptions 1.1, 1.2 and Algorithm 1 generate a sequence $\{w_t\}_{t \geq 0}$. Then we have for any $t \geq 0$ and $\eta$*

$$
P(w_{t+1}) - P^* \leq \left( 1 - \frac{\eta \mu}{\Gamma} \right) (P(w_t) - P^*) + \frac{\eta}{2\alpha} \|\nabla P(w_t) - v_t\|^2 - \left( \frac{1}{2\eta} - \frac{L}{2\alpha} \right) \|w_{t+1} - w_t\|_{\hat{D}_t}^2. \tag{10}
$$

**Proof:** According Lemma E.1, we have

$$
P(w_{t+1}) \leq P(w_t) + \frac{\eta}{2\alpha} \|\nabla P(w_t) - v_t\|^2 - \left( \frac{1}{2\eta} - \frac{L}{2\alpha} \right) \|w_{t+1} - w_t\|_{\hat{D}_t}^2 - \frac{\eta}{2} \|\nabla P(w_t)\|_{\hat{D}_t^{-1}}^2.
$$

Then with $\hat{D}_t^{-1} \preccurlyeq \frac{1}{\Gamma} I$ and PL-condition, we have

$$
\begin{aligned}
P(w_{t+1}) - P^* \leq & P(w_t) - P^* + \frac{\eta}{2\alpha} \|\nabla P(w_t) - v_t\|^2 - \left( \frac{1}{2\eta} - \frac{L}{2\alpha} \right) \|w_{t+1} - w_t\|_{\hat{D}_t}^2 \\
& - \frac{\eta}{2\Gamma} \|\nabla P(w_t)\|^2 \\
\leq & P(w_t) - P^* + \frac{\eta}{2\alpha} \|\nabla P(w_t) - v_t\|^2 - \left( \frac{1}{2\eta} - \frac{L}{2\alpha} \right) \|w_{t+1} - w_t\|_{\hat{D}_t}^2 \\
& - \frac{\eta \mu}{\Gamma} (P(w_t) - P^*).
\end{aligned}
$$

$\square$

**Lemma E.3.** *Suppose that Assumption 1.1 holds. Then we have*

$$
\mathbb{E} \left[ \|v_{t+1} - \nabla P(w_{t+1})\|^2 \right] \leq (1-p) \mathbb{E} \left[ \|v_t - \nabla P(w_t)\|^2 \right] + \frac{(1-p)L^2}{\alpha} \mathbb{E} \left[ \|w_{t+1} - w_t\|_{\hat{D}_t}^2 \right].
$$

**Proof:** Lemma 3 from (Li et al., 2021a) gives

$$
\mathbb{E} \left[ \|v_{t+1} - \nabla P(w_{t+1})\|^2 \right] \leq (1-p) \mathbb{E} \left[ \|v_t - \nabla P(w_t)\|^2 \right] + (1-p)L^2 \mathbb{E} \left[ \|w_{t+1} - w_t\|^2 \right].
$$

Using $I \preccurlyeq \frac{1}{\alpha} \hat{D}_t$, we get

$$
\mathbb{E} \left[ \|v_{t+1} - \nabla P(w_{t+1})\|^2 \right] \leq (1-p) \mathbb{E} \left[ \|v_t - \nabla P(w_t)\|^2 \right] + \frac{(1-p)L^2}{\alpha} \mathbb{E} \left[ \|w_{t+1} - w_t\|_{\hat{D}_t}^2 \right].
$$

$\square$

**Theorem E.4** (Theorem 3.1). *Suppose that Assumption 1.1 holds, let $\epsilon > 0$, let $p$ denote the probability, and let the step-size satisfy*

$$
\eta \leq \frac{\alpha}{L \left( 1 + \sqrt{\frac{1-p}{p}} \right)}.
$$

*Then, the number of iterations performed by* `Scaled SARAH`, *starting from an initial point $w_0 \in \mathbb{R}^d$ with $\Delta_0 = P(w_0) - P^*$, required to obtain an $\varepsilon$-approximate solution of the non-convex finite-sum problem* (1) *can be bounded by*

$$T = \mathcal{O}\left(\frac{\Gamma}{\alpha}\frac{\Delta_0 L}{\varepsilon^2}\left(1 + \sqrt{\frac{1-p}{p}}\right)\right).$$

**Proof:** Using Lemmas E.1, E.3, we have

$$\mathbb{E}\left[P(w_{t+1}) - P^* + \frac{\eta}{2\alpha p}\|\nabla P(w_{t+1}) - v_{t+1}\|^2\right]$$

$$\leq \mathbb{E}\left[P(w_t) - P^* + \frac{\eta}{2\alpha}\|\nabla P(w_t) - v_t\|^2\right]$$

$$- \left(\frac{1}{2\eta} - \frac{L}{2\alpha}\right)\mathbb{E}\left[\|w_{t+1} - w_t\|^2_{\hat{D}_t}\right] - \frac{\eta}{2}\mathbb{E}\left[\|\nabla P(w_t)\|^2_{\hat{D}_t^{-1}}\right]$$

$$+ \frac{\eta}{2\alpha p}\left((1-p)\mathbb{E}\left[\|v_t - \nabla P(w_t)\|^2\right] + \frac{(1-p)L^2}{\alpha}\mathbb{E}\left[\|w_{t+1} - w_t\|^2\right]\right)$$

$$= \mathbb{E}\left[P(w_t) - P^* + \frac{\eta}{2\alpha p}\|\nabla P(w_t) - v_t\|^2\right] - \frac{\eta}{2}\mathbb{E}\left[\|\nabla P(w_t)\|^2_{\hat{D}_t^{-1}}\right]$$

$$- \left(\frac{1}{2\eta} - \frac{L}{2\alpha} - \frac{(1-p)L^2}{2\alpha^2 p}\eta\right)\mathbb{E}\left[\|w_{t+1} - w_t\|^2\right].$$

Choosing $\eta \leq \frac{\alpha}{L\left(1+\sqrt{\frac{1-p}{p}}\right)}$ and defining $\Psi_{t+1} = P(w_{t+1}) - P^* + \frac{\eta}{2\alpha p}\|\nabla P(w_{t+1}) - v_{t+1}\|^2$, we have

$$\mathbb{E}\left[\Psi_{t+1}\right] \leq \mathbb{E}\left[\Psi_t\right] - \frac{\eta}{2}\mathbb{E}\left[\|\nabla P(w_t)\|^2_{\hat{D}_t^{-1}}\right].$$

Summing up, we obtain

$$\sum_{t=0}^{T-1}\frac{\eta}{2}\mathbb{E}\left[\|\nabla P(w_t)\|^2_{\hat{D}_t^{-1}}\right] \leq \mathbb{E}\left[\Psi_0\right] - \mathbb{E}\left[\Psi_T\right].$$

Using that $\hat{w}_T$ is chosen uniformly from all $w_t$ from 0 to $T-1$, we have

$$\frac{\eta}{2}\mathbb{E}\left[\|\nabla P(\hat{w}_T)\|^2_{\hat{D}_t^{-1}}\right] \leq \frac{\mathbb{E}\left[\Psi_0\right] - \mathbb{E}\left[\Psi_T\right]}{T}.$$

With $\Delta_0 = \Psi_0 = P(w_0) - P^*$, we get

$$\mathbb{E}\|\nabla P(\hat{w}_T)\|^2 \leq \Gamma\mathbb{E}\|\nabla P(\hat{w}_T)\|^2_{\hat{D}_t^{-1}} \leq \frac{2\Delta_0\Gamma}{\eta T}.$$

Then

$$T = \mathcal{O}\left(\frac{\Delta_0\Gamma}{\eta\varepsilon^2}\right) = \mathcal{O}\left(\frac{L\Delta_0}{\varepsilon^2}\frac{\Gamma}{\alpha}\sqrt{1 + \frac{1-p}{p}}\right).$$

$\square$

**Theorem E.5** (Theorem 3.2). *Suppose that Assumptions 1.1 and 1.2 hold, let $\epsilon > 0$, and let the step-size satisfy*

$$\eta \leq \frac{\alpha}{L\left(1 + \sqrt{\frac{1-p}{p}}\right)}.$$

*Then the number of iterations performed by* `Scaled SARAH` *sufficient for finding an $\varepsilon$-approximate solution of non-convex finite-sum problem* (1) *can be bounded by*

$$T = \mathcal{O}\left(\max\left\{\frac{1}{p}, \frac{L}{\mu}\frac{\Gamma}{\alpha}\left(1 + \sqrt{\frac{1-p}{p}}\right)\right\}\log\frac{\Delta_0}{\varepsilon}\right).$$

**Proof:** Using Lemmas E.2, E.3, we have for some $B > 0$

$$\mathbb{E}\left[P(w_{t+1}) - P^* + B\|\nabla P(w_{t+1}) - v_{t+1}\|^2\right]$$

$$\leq \left(1 - \frac{\eta\mu}{\Gamma}\right)\mathbb{E}\left[P(w_t) - P^*\right] + \frac{\eta}{2\alpha}\mathbb{E}\left[\|\nabla P(w_t) - v_t\|^2\right]$$

$$- \left(\frac{1}{2\eta} - \frac{L}{2\alpha}\right)\mathbb{E}\left[\|w_{t+1} - w_t\|_{\hat{D}_t}^2\right]$$

$$+ B\left((1-p)\mathbb{E}\left[\|v_t - \nabla P(w_t)\|^2\right] + \frac{(1-p)L^2}{\alpha}\mathbb{E}\left[\|w_{t+1} - w_t\|_{\hat{D}_t}^2\right]\right)$$

$$\leq \left(1 - \frac{\eta\mu}{\Gamma}\right)\mathbb{E}\left[P(w_t) - P^*\right] + \left(\frac{\eta}{2\alpha} + B(1-p)\right)\mathbb{E}\left[\|\nabla P(w_t) - v_t\|^2\right]$$

$$- \left(\frac{1}{2\eta} - \frac{L}{2\alpha} - \frac{(1-p)L^2}{\alpha}B\right)\mathbb{E}\left[\|w_{t+1} - w_t\|_{\hat{D}_t}^2\right].$$

We need to choose $\eta$, $B$ such that

$$\frac{\eta}{2\alpha} + B(1-p) \leq \left(1 - \frac{\eta\mu}{\Gamma}\right)B; \qquad \frac{1}{2\eta} - \frac{L}{2\alpha} - \frac{(1-p)L^2}{\alpha}B \geq 0.$$

If we take $B = \frac{\eta}{p\alpha}$ and $\eta \leq \min\{\frac{p\Gamma}{2\mu}, \frac{\alpha}{2L\left(1+\sqrt{\frac{1-p}{p}}\right)}\}$, we get

$$\mathbb{E}\left[P(w_{t+1}) - P^* + B\|\nabla P(w_{t+1}) - v_{t+1}\|^2\right] \leq \left(1 - \frac{\eta\mu}{\Gamma}\right)\mathbb{E}\left[P(w_t) - P^* + B\|\nabla P(w_t) - v_t\|^2\right].$$

and then obtain

$$\mathbb{E}\left[P(w_T) - P^* + B\|\nabla P(w_T) - v_T\|^2\right] \leq \left(1 - \frac{\eta\mu}{\Gamma}\right)^T \Delta_0.$$

Finally,

$$T = \mathcal{O}\left(\frac{\Gamma}{\eta\mu}\log\frac{\Delta_0}{\varepsilon}\right) = \mathcal{O}\left(\max\left\{\frac{1}{p}, \frac{L}{\mu}\frac{\Gamma}{\alpha}\left(1 + \sqrt{\frac{1-p}{p}}\right)\right\}\log\frac{\Delta_0}{\varepsilon}\right).$$

$\square$

# F  Scaled L-SVRG

Here we provide proofs for our theoretical results for `Scaled L-SVRG`.

**Lemma F.1.** *Suppose that Algorithm 2 generate sequences $\{w_t\}_{t\geq 0}$ and $\{z_t\}_{t\geq 0}$. Then we have for any $t \geq 0$, any $\eta$ and $B > 0$*

$$\mathbb{E}_t\left[\|w_{t+1} - z_{t+1}\|^2\right] \leq \frac{\eta^2}{\alpha}\mathbb{E}_t\left[\|v_t\|_{\hat{D}_t^{-1}}^2\right] + (1-p)(1+\eta B)\left[\|w_t - z_t\|^2\right] + (1-p)\frac{\eta}{\alpha B}\|\nabla P(w_t)\|_{\hat{D}_t^{-1}}^2.$$

**Proof:** Using definition $z_{t+1}$ and an update of `Scaled L-SVRG`: $w_{t+1} = w_t - \eta\hat{D}_t^{-1}v_t$, we have

$$\mathbb{E}\left[\mathbb{E}_{z_{t+1}}\left[\|w_{t+1} - z_{t+1}\|^2\right]\right] = p\mathbb{E}\left[\|w_{t+1} - w_t\|^2\right] + (1-p)\mathbb{E}\left[\|w_{t+1} - z_t\|^2\right]$$

$$= \frac{p\eta^2}{\alpha}\mathbb{E}\left[\|v_t\|_{\hat{D}_t^{-1}}^2\right] + (1-p)\mathbb{E}\left[\|w_{t+1} - z_t\|^2\right].$$

Here we additionally use $\hat{D}_t^{-1} \preccurlyeq \frac{1}{\alpha}I$. Next, we estimate $\mathbb{E}\left[\|w_{t+1} - z_t\|^2\right]$:

$$\mathbb{E}\left[\|w_{t+1} - z_t\|^2\right] = \mathbb{E}\left[\|w_t - \eta\hat{D}_t^{-1}v_t - z_t\|^2\right]$$

$$= \mathbb{E}\left[\|w_t - z_t\|^2\right] + \frac{\eta^2}{\alpha}\mathbb{E}\left[\|v_t\|_{\hat{D}_t^{-1}}^2\right] - 2\eta\mathbb{E}\left[\langle\hat{D}_t^{-1}v_t, w_t - z_t\rangle\right].$$

Using unbiasedness of $v_t$, we get for some $B > 0$

$$\mathbb{E}\left[\|w_{t+1} - z_t\|^2\right] = \mathbb{E}\left[\|w_t - z_t\|^2\right] + \frac{\eta^2}{\alpha}\mathbb{E}\left[\|v_t\|^2_{\hat{D}_t^{-1}}\right] - 2\eta\mathbb{E}\left[\langle \hat{D}_t^{-1}\nabla P(w_t), w_t - z_t\rangle\right]$$

$$\leq \mathbb{E}\left[\|w_t - z_t\|^2\right] + \frac{\eta^2}{\alpha}\mathbb{E}\left[\|v_t\|^2_{\hat{D}_t^{-1}}\right]$$

$$+ \eta\left(\frac{1}{\alpha B}\mathbb{E}\left[\|\nabla P(w_t)\|^2_{\hat{D}_t^{-1}}\right] + B\mathbb{E}\left[\|w_t - z_t\|^2\right]\right)$$

$$= (1 + \eta B)\mathbb{E}\left[\|w_t - z_t\|^2\right] + \frac{\eta}{\alpha B}\mathbb{E}\left[\|\nabla P(w_t)\|^2_{\hat{D}_t^{-1}}\right] + \frac{\eta^2}{\alpha}\mathbb{E}\left[\|v_t\|^2_{\hat{D}_t^{-1}}\right].$$

Combining inequality and equation, we finish proof.

$\square$

**Lemma F.2.** *Suppose that function $P$ satisfies Assumption 1.1 and Algorithm 2 generate a sequence $\{w_t\}_{t\geq 0}$. Then we have for any $t \geq 0$ and $\eta$*

$$\mathbb{E}\left[\|v_t\|^2_{\hat{D}_t^{-1}}\right] \leq 3\mathbb{E}\left[\|\nabla P(w_t)\|^2_{\hat{D}_t^{-1}}\right] + \frac{6L^2}{\alpha}\mathbb{E}\left[\|w_t - z_t\|^2\right]. \tag{11}$$

**Proof:** Using definition of $v_t$, we have

$$\mathbb{E}\left[\|v_t\|^2_{\hat{D}_t^{-1}}\right] = \mathbb{E}\left[\|\nabla f_{i_t}(w_t) - \nabla f_{i_t}(z_t) + \nabla P(z_t)\|^2_{\hat{D}_t^{-1}}\right]$$

$$\leq 3\mathbb{E}\left[\|\nabla P(w_t)\|^2_{\hat{D}_t^{-1}}\right] + 3\mathbb{E}\left[\|\nabla P(w_t) - \nabla P(z_t)\|^2_{\hat{D}_t^{-1}}\right]$$

$$+ 3\mathbb{E}\left[\|\nabla f_{i_t}(w_t) - \nabla f_{i_t}(z_t)\|^2_{\hat{D}_t^{-1}}\right]$$

$$\leq 3\mathbb{E}\left[\|\nabla P(w_t)\|^2_{\hat{D}_t}\right] + \frac{6L^2}{\alpha}\mathbb{E}\left[\|w_t - z_t\|^2\right].$$

Here we use Assumption 1.1 and $\hat{D}_t^{-1} \preccurlyeq \frac{1}{\alpha}I$.

$\square$

**Theorem F.3** (Theorem 3.1). *Suppose that Assumption 1.1 holds, let $\epsilon > 0$, let $p$ denote the probability and let the step-size satisfy*

$$\eta \leq \min\left\{\frac{\alpha}{4L}, \frac{\sqrt{p}\alpha}{\sqrt{24}L}, \frac{p^{2/3}}{144^{2/3}}\frac{\alpha}{L}\right\}.$$

*Given an initial point $w_0 \in \mathbb{R}^d$, let $\Delta_0 = P(w_0) - P^*$. Then the number of iterations performed by* Scaled L-SVRG, *starting from $w_0$, required to obtain an $\varepsilon$-approximate solution of non-convex finite-sum problem (1) can be bounded by*

$$T = \mathcal{O}\left(\frac{\Gamma}{\alpha}\frac{L\Delta_0}{p^{2/3}\varepsilon^2}\right).$$

**Proof:** Using $L$-smoothness of the function $P$ and $I \preccurlyeq \frac{1}{\alpha}\hat{D}_t$, we have

$$\mathbb{E}\left[P(w_{t+1})\right] \leq \mathbb{E}\left[P(w_t)\right] + \mathbb{E}\left[\langle \nabla P(w_t), w_{t+1} - w_t\rangle\right] + \frac{L}{2}\mathbb{E}\left[\|w_{t+1} - w_t\|^2\right]$$

$$\leq \mathbb{E}\left[P(w_t)\right] + \mathbb{E}\left[\langle \nabla P(w_t), w_{t+1} - w_t\rangle\right] + \frac{L}{2\alpha}\mathbb{E}\left[\|w_{t+1} - w_t\|^2_{\hat{D}_t}\right].$$

Taking into account an update of Scaled L-SVRG, we obtain

$$\mathbb{E}\left[P(w_{t+1}) - P^*\right] \leq \mathbb{E}\left[P(w_t) - P^*\right] + \mathbb{E}\left[\langle \nabla P(w_t), -\eta\hat{D}_t^{-1}v_t\rangle\right] + \frac{L\eta^2}{2\alpha}\mathbb{E}\left[\|v_t\|^2_{\hat{D}_t^{-1}}\right]$$

$$= \mathbb{E}\left[P(w_t) - P^*\right] - \eta\mathbb{E}\left[\langle \nabla P(w_t), \hat{D}_t^{-1}\nabla P(w_t)\rangle\right] + \frac{L\eta^2}{2\alpha}\mathbb{E}\left[\|v_t\|^2_{\hat{D}_t^{-1}}\right].$$

Let us define $\Phi_{t+1} = P(w_{t+1}) - P^* + A\|w_{t+1} - z_{t+1}\|^2$ for some $A > 0$. Using Lemmas F.1, F.2, we have

$$
\begin{aligned}
\mathbb{E}\left[\Phi_{t+1}\right] \leq & \mathbb{E}\left[P(w_t) - P^*\right] - \eta \mathbb{E}\left[\|\nabla P(w_t)\|^2_{\hat{D}_t^{-1}}\right] + \frac{L\eta^2}{2\alpha}\mathbb{E}\left[\|v_t\|^2_{\hat{D}_t^{-1}}\right] + A\mathbb{E}\left[\|w_{t+1} - z_{t+1}\|^2\right] \\
\leq & \mathbb{E}\left[P(w_t) - P^*\right] - \eta\mathbb{E}\left[\|\nabla P(w_t)\|^2_{\hat{D}_t^{-1}}\right] + \eta^2\left(\frac{L}{2\alpha} + \frac{A}{\alpha}\right)\mathbb{E}\left[\|v_t\|^2_{\hat{D}_t^{-1}}\right] \\
& + A(1-p)(1 + \eta B)\mathbb{E}\left[\|w_t - z_t\|^2\right] + A(1-p)\frac{\eta}{\alpha B}\mathbb{E}\left[\|\nabla P(w_t)\|^2_{\hat{D}_t^{-1}}\right] \\
\leq & \mathbb{E}\left[P(w_t) - P^*\right] - \eta\left(1 - \frac{A(1-p)}{\alpha B}\right)\mathbb{E}\left[\|\nabla P(w_t)\|^2_{\hat{D}_t^{-1}}\right] \\
& + A(1-p)(1 + \eta B)\mathbb{E}\left[\|w_t - z_t\|^2\right] \\
& + \eta^2\left(\frac{L}{2\alpha} + \frac{A}{\alpha}\right)\left(2\mathbb{E}\left[\|\nabla P(w_t)\|^2_{\hat{D}_t^{-1}}\right] + \frac{2L^2}{\alpha}\mathbb{E}\left[\|w_t - z_t\|^2\right]\right) \\
\leq & \mathbb{E}\left[P(w_t) - P^*\right] - \eta\left(1 - \frac{A(1-p)}{\alpha B} - \frac{2A}{\alpha}\eta - \frac{L}{\alpha}\eta\right)\mathbb{E}\left[\|\nabla P(w_t)\|^2_{\hat{D}_t^{-1}}\right] \\
& + A\left((1-p)(1 + \eta B) + \eta^2\left(\frac{L}{A} + 2\right)\frac{L^2}{\alpha^2}\right)\mathbb{E}\left[\|w_t - z_t\|^2\right].
\end{aligned}
\tag{12}
$$

We need to choose $A$, $\eta$, $B$ in such way:

$$
1 - \frac{A(1-p)}{\alpha B} - \frac{2A}{\alpha}\eta - \frac{L}{\alpha}\eta \geq \frac{1}{4}; \quad (1-p)(1 + \eta B) + \eta^2\left(\frac{L}{A} + 2\right)\frac{L^2}{\alpha^2} \leq 1.
$$

Thus, taking

$$
A = \frac{3\eta^2 L^3}{p\alpha^2}; \quad B = \frac{p}{3\eta}; \quad \eta \leq \min\left\{\frac{1}{4}, \left(\frac{p}{6}\right)^{1/2}, \left(\frac{p}{6}\right)^{2/3}\right\}\frac{\alpha}{L},
$$

we have

$$
\mathbb{E}\left[\Phi_{t+1}\right] \leq \mathbb{E}\left[\Phi_t\right] - \frac{\eta}{4}\mathbb{E}\left[\|\nabla P(w_t)\|^2_{\hat{D}_t^{-1}}\right].
$$

Summing up and using that $\hat{w}_T$ is chosen uniformly from all $w_t$ from 0 to $T - 1$, we get

$$
\mathbb{E}\|\nabla P(\hat{w}_T)\|^2 \leq \Gamma\mathbb{E}\|\nabla P(\hat{w}_t)\|^2_{\hat{D}_t^{-1}} \leq \frac{4\Delta_0\Gamma}{T\eta} \leq \max\left\{1, \sqrt{\frac{6}{p}}, \frac{\sqrt[3]{36}}{p^{2/3}}\right\}\frac{4L\Delta_0}{T}\frac{\Gamma}{\alpha}.
$$

Then

$$
T = \mathcal{O}\left(\frac{\Gamma}{\alpha}\frac{L\Delta_0}{p^{2/3}\varepsilon^2}\right).
$$

$\square$

**Theorem F.4** (Theorem A.2). *Suppose that Assumptions 1.1 and 1.2 hold, let $\epsilon > 0$, let $p$ denote the probability and let the step-size satisfy*

$$
\eta \leq \min\left\{\frac{p\Gamma}{6\mu}, \frac{1}{4}\frac{\alpha}{L}, \left(\frac{p}{6}\right)^{1/2}\frac{\alpha}{L}, \left(\frac{p}{6}\right)^{2/3}\frac{\alpha}{L}\right\}.
$$

*Then the number of iterations performed by* Scaled L-SVRG *sufficient for finding an $\varepsilon$-approximate solution of non-convex finite-sum problem (1) can be bounded by*

$$
T = \mathcal{O}\left(\max\left\{\frac{1}{p}, \frac{\Gamma}{\alpha}\frac{L}{p^{2/3}\mu}\right\}\right).
$$

**Proof:** Starting from (12), we get

$$
\begin{aligned}
\mathbb{E}\left[\Phi_{t+1}\right] \leq & \mathbb{E}\left[P(w_t) - P^*\right] - \eta\left(1 - \frac{A(1-p)}{\alpha B} - \frac{2A}{\alpha}\eta - \frac{L}{\alpha}\eta\right)\mathbb{E}\left[\|\nabla P(w_t)\|^2_{\hat{D}_t^{-1}}\right] \\
& + A\left((1-p)(1 + \eta B) + \eta^2\left(\frac{L}{A} + 2\right)\frac{L^2}{\alpha^2}\right)\mathbb{E}\left[\|w_t - z_t\|^2\right].
\end{aligned}
$$

We use $\hat{D}_t^{-1} \preccurlyeq \frac{1}{\Gamma}I$ and Assumption 1.2 and then get

$$
\begin{aligned}
\mathbb{E}\left[\Phi_{t+1}\right] \leq & \mathbb{E}\left[P(w_t) - P^*\right] - \frac{\eta}{\Gamma}\left(1 - \frac{A(1-p)}{\alpha B} - \frac{2A}{\alpha}\eta - \frac{L}{\alpha}\eta\right)\mathbb{E}\left[\|\nabla P(w_t)\|^2\right] \\
& + A\left((1-p)(1+\eta B) + \eta^2\left(\frac{L}{A}+2\right)\frac{L^2}{\alpha^2}\right)\mathbb{E}\left[\|w_t - z_t\|^2\right] \\
\leq & \mathbb{E}\left[P(w_t) - P^*\right] - \frac{\eta\mu}{\Gamma}\left(1 - \frac{A(1-p)}{\alpha B} - \frac{2A}{\alpha}\eta - \frac{L}{\alpha}\eta\right)\mathbb{E}\left[P(w_t) - P^*\right] \\
& + A\left((1-p)(1+\eta B) + \eta^2\left(\frac{L}{A}+2\right)\frac{L^2}{\alpha^2}\right)\mathbb{E}\left[\|w_t - z_t\|^2\right].
\end{aligned}
$$

We need to choose $A$, $\eta$, $B$ in such way:

$$
1 - \frac{A(1-p)}{\alpha\beta} - \frac{2A}{\alpha}\eta - \frac{L}{\alpha}\eta \geq \frac{1}{4}; \quad (1-p)(1+\eta\beta) + \eta^2\left(\frac{L}{A}+2\right)\frac{L^2}{\alpha^2} \leq 1 - \frac{\mu\eta}{4\Gamma}.
$$

Thus, taking

$$
A = \frac{3\eta^2 L^3}{p\alpha^2}; \quad \beta = \frac{p}{3\eta}; \quad \eta \leq \min\left\{\frac{p\Gamma}{6\mu}, \frac{1}{4}\frac{\alpha}{L}, \left(\frac{p}{6}\right)^{1/2}\frac{\alpha}{L}, \left(\frac{p}{6}\right)^{2/3}\frac{\alpha}{L}\right\},
$$

we have

$$
\mathbb{E}\left[\Phi_{t+1}\right] \leq \left(1 - \min\left\{\frac{p}{24}, \frac{1}{16}\frac{\alpha\mu}{\Gamma L}, \left(\frac{p}{6}\right)^{1/2}\frac{\alpha\mu}{4\Gamma L}, \left(\frac{p}{6}\right)^{2/3}\frac{\alpha\mu}{4\Gamma L}\right\}\right)\mathbb{E}\left[\Phi_t\right].
$$

Then

$$
T = \mathcal{O}\left(\max\left\{\frac{1}{p}, \frac{\Gamma}{\alpha}\frac{L}{p^{2/3}\mu}\right\}\right).
$$

## G  THE ROLE OF $\beta$ IN THE PRECONDITIONER

The purpose of this section is to better understand the role that $\beta$ plays with regards to the convergence theory and practical performance of our algorithms. Recall that $\beta$ is the momentum parameter for the preconditioner (3), and it controls the weighting/trade-off between the past curvature history and the current minibatch Hessian.

Note that the analysis presented in Appendices E and F does not impose any additional assumptions on $\beta$, which demonstrates a kind of universality of our method and convergence theory. Note that, Adam (Défossez et al., 2020) and OASIS do not have such universality. In particular, for Adam, the parameter $\beta_2$ (corresponding to $\beta$ in this work) is critical: for Adam to converge at the rate $1/\sqrt{T}$ in the non-convex setting, one must choose $\beta_2 = 1 - 1/\sqrt{T}$, while for small $\beta_2$ the method diverges (Reddi et al., 2019). The dependence of OASIS on $\beta$ is only presented in the adaptive case, but this dependence can negatively impact, or destroy, convergence. This section aims to better understand the choice $\beta$.

Consider the following quadratic function:

$$
f(x, y) = \frac{L}{2}\left(\frac{x+y}{2}\right)^2 + \frac{\mu}{2}\left(\frac{x-y}{2}\right)^2.
$$

The corresponding Hessian for this function is

$$
\nabla^2 f(x, y) = \begin{pmatrix} \frac{L+\mu}{2} & \frac{L-\mu}{2} \\ \frac{L-\mu}{2} & \frac{L+\mu}{2} \end{pmatrix},
$$

and

$$
\texttt{diag}\left(z_t \odot \nabla^2 f z_t\right) = \begin{pmatrix} \frac{L+\mu}{2} + \frac{L-\mu}{2}z_t^1 z_t^2 & 0 \\ 0 & \frac{L+\mu}{2} + \frac{L-\mu}{2}z_t^1 z_t^2 \end{pmatrix},
$$

where $z_t$ is from a Radermacher distribution. One can note that with probability $\frac{1}{2}$ our approximation $\texttt{diag}\left(z_t \odot \nabla^2 f z_t\right)$ is

$$
\begin{pmatrix} L & 0 \\ 0 & L \end{pmatrix} \quad \text{or} \quad \begin{pmatrix} \mu & 0 \\ 0 & \mu \end{pmatrix}. \tag{13}
$$

### G.1 THE WORST CASE

Intuitively, with a good understanding of $\beta$, it should be possible to get better practical performance of the method. For example, our convergence analysis depends upon the factor $\Gamma/\alpha$ (recall Remark 2.1), and a good choice of $\beta$ can shrink this factor. However, here we present the worst case, and show that for the function described above, unfortunately, improvement is not possible. Choose $\alpha > \mu$, so that for $\hat{D}_t$ we have two possible matrices with probability $\frac{1}{2}$:

$$
\begin{pmatrix} L & 0 \\ 0 & L \end{pmatrix} \quad \text{or} \quad \begin{pmatrix} \alpha & 0 \\ 0 & \alpha \end{pmatrix}.
$$

The matrices above are scaled identity matrices, so that applying the preconditioner is equivalent to simply dividing the gradient by a constant. This scaling can dramatically change size of gradient especially in the early iterations, or when $\beta = 0$. In particular, when we work with the second matrix we change gradient by $1/\alpha$. Then it is really worth taking a step $\eta \sim \frac{\alpha}{L}$: $\alpha$ – impact of scaling, $L$ – original stepsize for GD type methods. Additionally, if we randomly work with the first matrix with $L$, this additional $L$ goes to the convergence of the method as $\Gamma$.

### G.2 HOW TO CHOOSE THE CONSTANT $\beta$

$D_t$ is a linear combination of identically distributed independent matrices (13). In such a situation, it is natural to consider reducing the variance of $D_t$. Note that

$$
D_t = \beta^t \texttt{diag}\left(z_0 \odot \nabla^2 f z_0\right) + \sum_{\tau=1}^{t} \beta^{t-\tau}(1-\beta)\texttt{diag}\left(z_\tau \odot \nabla^2 f z_\tau\right).
$$

and notice that, by (13), every $\texttt{diag}\left(z_\tau \odot \nabla^2 f z_\tau\right)$ has the same distribution. Then the variance of $D_t$ can be computed as

$$
\text{Var}[D_t] = \left[\beta^{2t} + \sum_{\tau=1}^{t} \beta^{2t-2\tau}(1-\beta)^2\right] C,
$$

where $C$ is some constant (that does not depend on $\beta$). Now, minimizing this expression w.r.t. $\beta \in [0; 1]$:

$$
\min_{\beta \in [0;1]} \left[\beta^{2t} + \sum_{\tau=1}^{t} \beta^{2t-2\tau}(1-\beta)^2\right].
$$

Using the optimality conditions one has:
$$
\beta^{2t-1}(2t - 1 + 2t\beta + 2) = 1.
$$

If we have a limit of iterations $T$ and if we want to minimize the variance of the final preconditioner, we can take

$$
\beta = \frac{1}{\sqrt[2T-1]{2T - 1 + 2T\beta + 2}} \sim \frac{1}{\sqrt[2T]{2T}} \xrightarrow[T\to\infty]{} 1.
$$

### G.3 IT IS BETTER TO CONSIDER VARYING $\beta_t$

As in the previous section, here we also want to reduce the variance of $D_t$. But now, let us consider the case when $\beta_t$ is allowed to vary:

$$
D_t = \beta_t D_{t-1} + (1 - \beta_t)\texttt{diag}\left(z_t \odot \nabla^2 f z_t\right).
$$

To understand how to choose $\beta_t$, note that at each iteration, because $\texttt{diag}\left(z_t \odot \nabla^2 f z_t\right)$ we get one of the two matrices given in (13). To reduce the variance of the matrix $D_t$ one can choose $D_t$ as

$$D_t = \frac{1}{t+1}\sum_{\tau=0}^{t} \texttt{diag}\left(z_\tau \odot \nabla^2 f z_\tau\right) = \frac{t}{t+1}D_{t-1} + \frac{1}{t+1}\texttt{diag}\left(z_t \odot \nabla^2 f z_t\right).$$

Thus,

$$\beta_t = 1 - \frac{1}{t+1}.$$

Using the identity and independence of the $\texttt{diag}\left(z_\tau \odot \nabla^2 f z_\tau\right)$ distributions, one can note that such a $\beta_t$ gives better variance then the constant $\beta$ from the previous subsection.

## H    DRAFT OF EXPERIMENTS

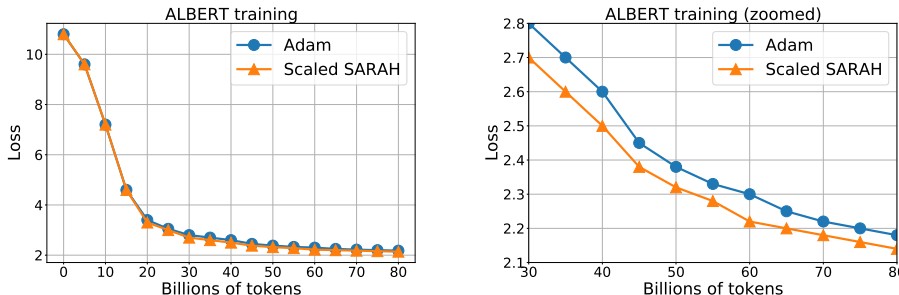

Figure 20: ALBERT training.

