# OpenReview forum: "Stochastic Gradient Methods with Preconditioned Updates"
_ICLR.cc/2023/Conference — Submitted to ICLR 2023_

### Official Review · Reviewer_QfWd · 2022-10-21

**Confidence:** 3
**Correctness:** 3
**Technical Novelty And Significance:** 3
**Empirical Novelty And Significance:** Not applicable
**Recommendation:** 5

**Clarity, Quality, Novelty And Reproducibility:**

The paper is written quite clearly.
The novelty is mainly in the convergence bounds --- the algorithms themselves combine a previously known technique with two known algorithms.

**Strength And Weaknesses:**

The paper combines two well known algorithms with a previous preconditioner. The key theoretical result (which I did not check in detail) is that the preconditioned algorithms have a better convergence bound than previous algorithms such as AdaGrad, Adam etc.

Points to improve:
1. Please compare the convergence bounds of the Scaled algorithms with the unscaled versions. Also for Adagrad and Adam, please do point out the specific theorems in the papers where the epsilon^{-4} bounds are shown.
2. Some discussion of the importance of these bounds should be included. While the given algorithms are theoretically much more efficient than AdaGrad/Adam, the difference is not visible in the experimental results.
3. The experiments are on very small datasets. The big advantage of Adagrad and Adam is that they run on very large datasets, it would be definitely desirable to see the performance of Scaled SARAH and Scaled L-SVRG on some moderate sized datasets.

**Summary Of The Paper:**

This paper enhances two previous algorithms for non-convex finite sum minimization, SARAH and L-SVRG, by adding a diagonal preconditioner from Jahani et al. The authors prove that the preconditioned methods, which they call Scaled SARAH and Scaled L-SVRG, both converge. The convergence bounds are better than those of previous algorithms.
The authors then show the results of their methods on several LibSVM datasets.

**Summary Of The Review:**

I would like to see the weaknesses in the paper corrected before it is ready for publication. As of now, it makes strong claims, but they are not backed up by enough discussion of their significance, or by experimental results.

---

### Official Review · Reviewer_ayos · 2022-10-23

**Confidence:** 4
**Correctness:** 3
**Technical Novelty And Significance:** 2
**Empirical Novelty And Significance:** 1
**Recommendation:** 5

**Clarity, Quality, Novelty And Reproducibility:**

This paper presents very clearly. But the originality is limited, which still needs to be strengthened.

**Strength And Weaknesses:**

Strength: introducing a new method to improve the convergence of the stochastic method

Weaknesses: (1) The theoretical complexity is only for the smooth objective function.

                       (2) The function under the PL condition is equivalent to the strongly convex function, but the linear convergence   from Nesterov's method is classical.

**Summary Of The Paper:**

This paper proposes a preconditioned update to stochastic gradient methods， based upon Hutchinson’s approach to approximating the diagonal of the Hessian.

**Summary Of The Review:**

The paper proposes a preconditioned update to stochastic gradient methods, but the theoretical results still need to be strengthened.

---

### Official Review · Reviewer_8yK7 · 2022-10-23

**Confidence:** 3
**Correctness:** 3
**Technical Novelty And Significance:** 1
**Empirical Novelty And Significance:** 3
**Recommendation:** 5

**Clarity, Quality, Novelty And Reproducibility:**

I think the clarity and quality of writing is good. See above for my additional comments.

**Strength And Weaknesses:**

Strength:
1) I think the paper is well-presented. The authors explain the motivation and idea pretty clearly, thus making it easy to follow. The author also did a thorough literature review.

2) I think the problem they study is interesting and of important: i.e. study the effect of preconditioning, or incorporating little second-order information for finite-sum minimization-type methods. This is an important topic.

3)The empirical performance show favorable improvement over other SOTA optimizer like Adam, which demonstrates potential impact of this preconditioning idea to other mainstream first-order optimizers in training neural networks.

Weakness:
1) The main confusion I have is I don't think the theoretical results in this paper corroborate the main argument of the paper, i.e. pre-conditioning technique is useful in practice. In particular, the theoretical analysis with preconditioning all just suffer from an additional L/alpha in their runtimes (compared with standard SARAH and SVRG), which is basically the condition number in worst case of the approximate diagonal matrix you construct, right? This is not a very interesting theoretical result in my mind. In particular, it doesn't explain why you choose such an estimator for diagonal rescaling at all. I believe for any arbitrary diagonal preconditioning bounded by condition number L/alpha, you can prove the same result. Please correct me if I am wrong.

2) Following the above point, I feel the claim in the introduction about showing an eps^{-2} theoretical result, which improve over eps^{-4} is a bit misleading. Really you probably want to compare with un-preconditioned SARAH and other methods. The eps^{-4} result is for general case not finite sum minimization, and they also don't scale additionally with the number of function n, right? (please correct me if I am mistaken)

2) In its current way written, it is hard to compare this paper's theoretical result with some prior art. In particular in these settings a natural metric is the gradient complexity so adding a remark for Theorem 3.1, 3.2 etc. on that would be helpful. Also, it seems Theorem 3.1 right now doesn't require an additive n in the final bound? Is that possible, or in other words would this violate some lower bound?

**Summary Of The Paper:**

The paper provides a generic pre-conditioning method for non-convex finite-sum minimization, which is an important task in machine learning. It combines this technique with two mainstream algorithms for finite-sum minimization, namely SARAH and L-SVRG. They provide both theoretical analysis for these algorithms, showing it achieves 1/eps^2 convergence rate under smoothness condition and log(1/eps) convergence rate under PL condition. They also corroborate the favorable performance of this pre-conditioning technique in practice by comparing it with Adam, another algorithm that incorporates second order information adaptively.

**Summary Of The Review:**

I like the main idea of the paper a lot. And I think the experiment performance looks quite interesting too.

I just feel theoretical analysis could probably be improved, or if not the main contributions and comparison with prior art shall be made clearer in the paper. Otherwise the theory part of the paper really doesn't feel like making much point in the current form.

---

### Decision · Program_Chairs · 2023-01-20

**Decision:**

Reject

**Justification For Why Not Higher Score:**

As laid out in my meta review, the reviewers found the paper lacking in the novelty of ideas, more thorough and relevant experimentation and a significant advancement of the state of knowledge through its theoretical bounds.

**Justification For Why Not Lower Score:**

NA

**Metareview: Summary, Strengths And Weaknesses:**

The paper considers the problem of non-convex finite sum optimization and proposes diagonally preconditioned versions of existing variance reduction methods such as SARAH, SVRG. The authors via experiments on LibSVM datasets show an empirical advantage of making this change on these methods where the proposed algorithms show faster convergence. The paper provides theoretical convergence bounds for their proposed methods.

Overall the reviewers had the following main concerns regarding the paper -
1. The core idea of the paper is a combination of (well)-known ideas in literature. Variance reduction and diagonal preconditioning. They use a relatively more recent preconditioning method but has appeared in literature before as well. This by itself is not sufficient to lead to the decision but combined with the following factors makes the paper fall below the bar.
2. The reviewers found experimental evaluation to be limited. The paper provides experimentation on LibSVM datasets and in the rebuttal phase provided a single deep learning experiment, but to be fully relevant to the conference community the reviewers were of the opinion that the paper should provide a more detailed comparison of their algorithm against standard benchmarks. The single experiment provided by the paper in the rebuttal phase is unfortunately not sufficient and the reviewers understand that it is hard to provide strong such experiments in limited time. Therefore they recommend that the authors perform a more detailed comparison (also presenting it in detail) in the next version.
3. The theoretical bounds in the paper do not unfortunately provide a clear improvement in the SOTA. The reviewers understand and agree with the authors that 'such is the state of our understanding of adaptive methods from the lens of the convergence bounds'. However the reviewers also maintained that the paper unfortunately did not introduce new ideas from a theoretical perspective or the bounds presented in the paper did not do enough to expand the referred state of understanding.

Please note none of these factors have lead to the decision singularly but rather the combination of them. As such if any one aspect of the paper significantly outweighed the other demerits it would have been considered commensurately. As it stands the paper unfortunately falls below the bar for acceptance in the unanimous collective opinion of the reviewers and the meta reviewer.

The reviewers strongly suggested to do a more thorough comparison of the proposed method on more diverse settings including deep learning which is where diagonal preconditioners have shown sustained advantage/



**Summary Of Ac-Reviewer Meeting:**

The reviewers and the meta-reviewer met over VC and I have highlighted the points discussed in detail in my meta-review.